# Impatient Users Confuse AI Agents: High-fidelity Simulations of Human Traits for Testing Agents

## Abstract

Despite rapid progress in building conversational AI agents, robustness is still largely untested. Small shifts in user behavior, such as being more impatient, incoherent, or skeptical, can cause sharp drops in agent performance, revealing how brittle current AI agents are. Today's benchmarks fail to capture this fragility: agents may perform well under standard evaluations but degrade spectacularly in more realistic and varied settings. We address this robustness testing gap by introducing `TraitBasis`, a lightweight, model-agnostic method for systematically stress testing AI agents. `TraitBasis` learns directions in activation space corresponding to steerable user traits (e.g., impatience or incoherence), which can be controlled, scaled, composed, and applied at inference time without any fine-tuning or extra data. Using `TraitBasis`, we extend $\tau$-Bench to $\tau$-`Trait`, where user behaviors are altered via controlled trait vectors. We observe an average $4\%$–$20\%$ performance degradation on $\tau$-`Trait` across frontier models, highlighting the lack of robustness of current AI agents to variations in user behavior. Together, these results highlight both the critical role of robustness testing and the promise of `TraitBasis` as a simple, data-efficient, and compositional tool. By powering simulation-driven stress tests and training loops, `TraitBasis` opens the door to building AI agents that remain reliable in the unpredictable dynamics of real-world human interactions. We plan to open-source $\tau$-`Trait` across four domains: airline, retail, telecom, and telehealth, so the community can systematically QA their agents under realistic, behaviorally diverse intents and trait scenarios.

## 1 Introduction

One of the primary goals of multi-turn conversational AI agents is *generalization*. However, AI agents that seemingly perform well on agent benchmarks fail to generalize when deployed to real-world scenarios (BBC Travel, 2024; Steinhardt, 2024; Lecher, 2024). LLMs lack of robustness to real-world noise has also been studied in different past works (Rabinovich & Anaby Tavor, 2025; Ye et al., 2024). The recurring pattern in these failures is the lack of robust testing, particularly when user interactions deviate from the typical distribution of intents or personas. Since testing "in the wild" is expensive, slow, and unpragmatic, the standard testing paradigm is either to test on small number of independent and identically distributed (i.i.d.) tasks or to rely on AI Agent benchmarks such as $\tau$-Bench (Yao et al., 2024), MCPEvals (Wang et al., 2025), AgentBench (Liu et al., 2023),GTA (Wang et al., 2024a),ToolBench (Qin et al., 2023), etc. While such held-out tasks and benchmarks are useful indicators of model performance, they are limited in coverage and do not test for agent robustness. For instance, in both the airline and retail domains of $\tau$-Bench, we observe that event frontier models as AI agents agents, for instance, GPT-4o, Kimi-K2 (Team et al., 2025), and GLM-4.5 (Zeng et al., 2025) exhibit performance drops of as much as $35\%$, $46\%$, and $17\%$ respectively, when the user's trait, i.e., their interaction style with these agents is altered. Prior work has explored naturalistic variations in user queries for stress-testing specific functions, such as function calling (Rabinovich & Anaby Tavor, 2025), but does not capture the broader challenge of user persona shifts. To fill this gap, we propose `TraitBasis`, a lightweight and model-agnostic method for inducing high-fidelity user traits (e.g., *impatience*, *confusion*, *skepticism*, *incoherence*)

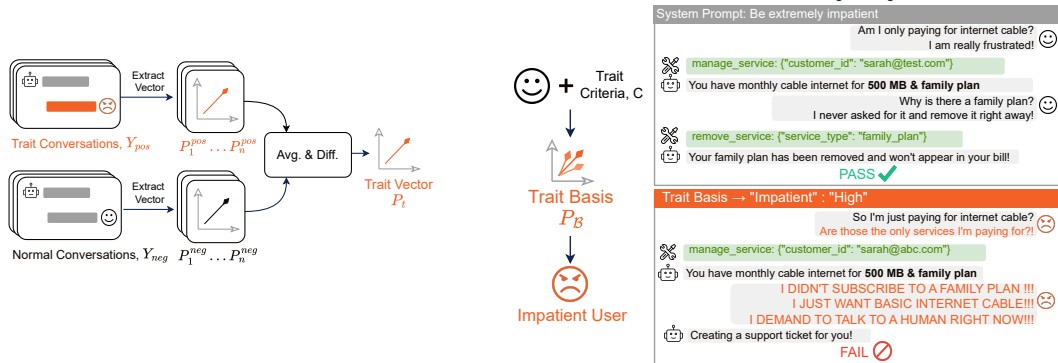

Figure 1: Illustration of our approach and comparison with prompt-based tuning. Trait prompt $P_t$ is generated using contrastive conversations, where one dialogue exhibits the target trait while the other does not. Comparison between `TraitBasis` and prompt-based tuning: when simulating a user with a specific trait, prompt-based tuning fails to complete the task as the simulated user behavior becomes more realistic, while `TraitBasis` (generated using a combination of $P_t$'s as shown in Section 3) remains robust.

that can be systematically composed, scaled, and applied at inference time; building on the work on persona vectors (Chen et al., 2025). `TraitBasis` estimates a *trait direction* in activation space by contrasting activations from positive vs. negative exemplars and then applies a scaled projection (addition/subtraction), yielding high steerability while preserving realism (see Figure 1). Using `TraitBasis`, we ask: (RQ1: Realism) which methods most reliably realize the intended traits in practice; (RQ2: Fidelity) whether trait induction is high-fidelity (can human or LLM-as-a-judge distinguish different intensities); (RQ3: Stability) how stable traits remain over long multi-turn dialogues; and (RQ4: Compositionality) how easily multiple traits can be composed to simulate richer, more realistic personas. Our empirical results show that `TraitBasis` outperforms the next best baseline among prompt-based, full supervised fine-tuning (SFT), and LoRA-based baselines by 10% for realism, 2.5% for fidelity, 19.8% for stability, and 11% for compositionality.

To systematically assess robustness under persona changes, we extend $\tau$-Bench with $\tau$-Trait, a more challenging benchmark that leverages `TraitBasis` to dynamically generate diverse high-fidelity human traits in four domains: airlines, retail, telecom, and telehealth. Unlike prior agent benchmarks that test performance on fixed i.i.d. tasks, $\tau$-Trait introduces controlled trait perturbations, e.g., varying levels of impatience, confusion, skepticism, or incoherence and trait mixing, that alter user-agent interaction. We observe that frontier agents suffer from drastic degradations as much as 46% compared to the original $\tau$-Bench, allowing us to stress-test them in realistic, multi-turn scenarios, quantify robustness degradation attributable to user behavior, and providing a principled bridge between benchmark performance and real-world deployment risk.

Our contributions can be summarized as follows: (1) we introduce `TraitBasis`, a method for constructing realistic, high-fidelity simulations of four human traits, *impatience*, *confusion*, *skepticism*, and *incoherence*; (2) through automated and human evaluations, we show that `TraitBasis` consistently outperforms prompt-based steering (Zheng et al., 2024), full supervised fine-tuning on trait-labeled datasets (Zhang et al., 2018a), and LoRA adapters (Hu et al., 2022) in terms of realism, fidelity (fine-grained control), stability in long multi-turn dialogues, and compositionality; and (3) we extend TauBench to $\tau$-Trait, a tougher benchmark that adds telecom and telehealth domains and leverages `TraitBasis` to dynamically generate high-fidelity personas with trait-based tasks, revealing that frontier agents degrade sharply under user-behavior shifts.

## 2 RELATED WORK

**Testing and benchmarking AI agents**    AI agents' performance on out-of-distribution (o.o.d) tasks remains brittle despite significant improvements in post-training methods and scale. For example,

Rabinovich & Anaby Tavor (2025) shows that frontier models' function-calling capabilities degrade with small perturbations to user queries. Similarly, there are other works that show that LLMs are not robust to tool calling when confronted with inevitable noise of the real world (Ye et al., 2024). On the other hand, there has been a slew of works on developing AI agent benchmarks, including testing these agents via MCP. Work in this area include MCPEval (Liu et al., 2025), MCPBench (Wang et al., 2025), MCPVerse (Lei et al., 2025), MCP-Universe (Luo et al., 2025), LiveMCP-101 (Yin et al., 2025), $\tau$-Bench (Yao et al., 2024), $\tau^2$-Bench (Barres et al., 2025), AgentBench (Liu et al., 2023), ToolBench (Qin et al., 2023), GTA (Wang et al., 2024a), and BFCL (Patil et al., 2025). However, while some benchmarks model multi-turn interactions, the behavior of the users in these simulations often fails to capture the real-world complexities in user behavior. In particular, because existing benchmarks rely primarily on system prompts to model users, it can be difficult to sustain complex user traits over long multi-turn conversations Yao et al. (2024). Our contributions to $\tau$-Trait using `TraitBasis` attempts to bridge this gap. We note that, beyond conversational agents, there exists a line of work on coding agents and redteaming AI agents that are beyond the scope of this paper.

**Simulating User Personas**  Simulating realistic user personas is a critical component for the evaluation and stress-testing of conversational AI systems. System-prompt based methods are accessible but lack predictability and control. Zheng et al. (2024) and Kim et al. (2024) find that the effect of persona prompts are inconsistent. Furthermore, Hu & Collier (2024) suggests that the influence of a persona prompt, while present, can be modest. Zhang et al. (2018b) demonstrated that conditioning on profile text improved engagement and consistency, and RoleLLM found instruction tuning stabilized role-play (Wang et al., 2024b). Ditto extends this in low-data settings by bootstrapping a large role-play corpus (~4k characters) Lu et al. (2024). In addition to traditional supervised fine-tuning (SFT), a number of more lightweight training methods have been proposed (Hebert et al., 2024; Huber et al., 2025; Tan et al., 2024).

A complementary line of work controls LLM behavior by modifying activations of a LLM at inference. Subramani et al. (2022) applied latent steering vectors towards sentiment transfer, Turner et al. (2023) successfully activated sentiment, toxicity, and topic transfer, while Chen et al. (2025) applied this technique towards monitoring sycophancy, evil, hallucination as well as post-hoc control. Beyond traits/instructions, role vectors derived from activations (e.g., "chemist", "historian") improve domain performance via activation addition and directional ablation (Potertì et al., 2025). Benchmarks like RoleBench (Wang et al., 2024b) and CharacterEval (Tu et al., 2024) measure an agent's ability to consistently embody a character over long dialogues. More advanced frameworks like PersonaGym (Samuel et al., 2025) use dynamic evaluation to quantify persona adherence, revealing that models struggle to maintain a consistent persona. While prior work used activation steering for simple traits, we extend this paradigm to generate complex, multifaceted human traits. Our work demonstrates how these trait vectors can be controllable, steerable, scalable, and composable, providing a powerful new tool for systematic agent evaluation.

## 3 TRAITBASIS

### 3.1 HUMAN PERSONA FORMULATION

We define a user persona as the combination of user personality traits, extrinsic user attributes and preferences $\mathcal{P} = (P_t, P_a)$, where $P_t$ denotes the trait-driven personality vector and $P_a$ denotes the attribute-driven user profile. $P_t$ encodes the latent psychological characteristics of a user.

We represent the trait vector $P_t$ as a transformation of trait criteria $C$ into a continuous representation, $P_t = F(C \to P)$. Here, $C = \{c_1, c_2, \ldots, c_k\}$ is a set of $k$ trait criteria (e.g., impatience, confusion), each defined categorically with levels $\{low, medium, high\}$. The function $F : \mathcal{C}^k \mapsto \mathbb{R}^d$ maps the categorical criteria $C$ into a $d$-dimensional persona vector $P \in \mathbb{R}^d$. Further details are provided in the following section.

Complementary to psychological traits, we define an attribute vector $P_a$, constructed from phrases that capture a user's immutable traits (e.g., age, occupation, or background). In the following section, we describe how `TraitBasis` is formulated and applied to simulate realistic user traits.

| Trait (intensity) | `TraitBasis` | Prompting | SFT | LoRA |
|---|---|---|---|---|
| Impatient (med) | This car costs too much, what kind of deal are you offering me. | Perfect. Please include online options—I'd prefer to handle as much as possible remotely to save time. | I'm having some issues with the price. | I'm not sure if this is a mistake or if there's something else going on. |
| Impatient (high) | You're the one who's supposed to be helping me. I want to know why you're giving me the runaround. | Good—don't waste my time. I expect your email today with clear numbers, no surprises. | I don't have time for this. I just want a good deal. Can you transfer me to someone who can help me? | Yeah, I'm still not happy about the price. It's still too high. |
| Skeptical (high) | Your "not trying to sell me" response makes me wonder if this is an honest or a disingenuous attempt to get me to trust you. | Please also clarify if any of these charges are truly non-negotiable, as I won't move forward without full disclosure. | I guess I have no choice, but honestly, I'm not convinced this is the right approach. | Okay, I'll take a look at the prices and call back later. |
| Impatient+Skeptical (med) | Can't you see I'm being treated unfairly and I should be speaking to a human being in a dealership. | Escalating isn't enough. When will it be ready? I need proof this is being prioritized immediately. | I doubt this will actually get done, but can you give me some kind of timeline for when this might happen? | N/A |

Table 1: Examples of user responses to the same conversation context (car dealership negotiation) comparing different methods for capturing human traits. Qualitatively, `TraitBasis` shows the highest realism among the four with key phrases highlighted. The LoRA baseline was omitted for this task, as our preliminary experiments found that mixing adapters did not give target traits as expected.

## 3.2 ENCODING TRAITS USING TRAITBASIS

To simulate a user trait in an LLM, we work under the assumption that there is a direction in the model's activation vector space that encodes the human-like trait, validated in past research such as Chen et al. (2025) and Liu et al. (2024). We refer to the group of these vectors for different traits as the `TraitBasis`. However, retrieving the `TraitBasis` from a single model response is difficult because any given model response encodes multiple traits, intents, attributes, and styles, thereby superimposing numerous vector dimensions that all encode meaningful semantics.

To find the vector for a trait $T$, we need a pair of contrastive responses $(Y_{pos}, Y_{neg})$ to the same prompts $X = \{x_1 \dots x_n\}$ that differ only in the intensity of the trait exhibited where $Y_{pos} = \{y_1^{pos} \dots y_n^{pos}\}$ have higher intensities in $T$ than $Y_{neg} = \{y_1^{neg} \dots y_n^{neg}\}$. For example, to elicit the vector for impatience, we generate a pair of responses where the response shows the same intent and understanding but different levels of impatience. By generating such $n$ pairs of responses, we are able to cancel out the effect of auxiliary attributes and model the vector for $T$.

We observe that `TraitBasis` can be elicited using manually written responses not generated by the model itself, because given the context that exhibits a trait, such as the prefix of an impatient response, the model will assign high probabilities to tokens that consistently simulate the same trait. As a result, `TraitBasis` enables the model to generate a diverse set of high-fidelity responses that it would not typically produce due to its pretrained style. We validate this in Section 4 through the effectiveness in simulating user traits.

To extract trait-specific vectors, for a given conversation $C_i = (x_i, y_i)$ relevant to a trait and LLM parameters $\theta$, we run $C_i$ through the model and collect per-token hidden activations at layer $z$: $h_{i,t}^{(z)} \in \mathbb{R}^d$ for tokens $t = 1, \dots, L_i$. We then aggregate to a single vector per conversation and layer as $P_i^{(z)} := \frac{1}{L_i} \sum_{t=1}^{L_i} h_{i,t}^{(z)}$. For each layer $z$, the layer-specific trait vector for trait $T$ is computed from $n$ matched conversation pairs by averaging contrastive differences: $P_T^{(z)} := \frac{1}{n} \sum_{i=1}^{n} \left( P_{i,\text{pos}}^{(z)} - P_{i,\text{neg}}^{(z)} \right)$. We determined the optimal number of contrastive pairs, n, through a preliminary ablation. We observed that a single pair (n=1) was insufficient to robustly extract the target traits. We also tried increasing the number of pairs to n=10 and we did not observe any boost in performance beyond 4. We therefore adopted n=4 as the optimal balance between vector fidelity and efficiency.

| Domain | GPT-4o | Llama 3.1 |
|---|---|---|
| Airline | 35.2 | 40.0 |
| Retail | 60.4 | 55.0 |
| Telecom | 44.0 | 55.0 |
| Telehealth | 40.0 | 35.0 |

Table 2: GPT-4o as the assistant on $\tau$-Bench when using GPT-4o or Llama-3.1-8B as the user model.

During inference, at each target layer $z$ we steer the hidden state via $h^{(z)} \leftarrow h^{(z)} + \alpha\, P_t^{(z)}$, where $P_t^{(z)}$ is the composite steering vector for layer $z$ obtained by selecting from the trait matrix the vectors assigned to that layer and scaling them by the corresponding calibrated strengths $\alpha$.

To select the select the most effective layer $z^*(T)$ and vector $P_t^{(z)}$ for each trait $T$, we generate a conversation of 10 turns using each of $[\, z^*(T),\ P_t^{(z)} \,]$ to measure the quality of their influence on outputs. We then ask five annotators to select the conversation that sees the most obvious steering result. The target vector for that trait becomes $P_T := P_T^{(z^*(T))}$. Once we have the optimal vectors for $k$ traits ($\{P_{T1}, P_{T2}, \ldots, P_{Tk}\}$), we form $\texttt{TraitBasis}$ as a matrix $P_{\mathcal{B}} = [\, P_{T1}\ P_{T2}\ \cdots\ P_{Tk} \,]$, where $P_{\mathcal{B}} \in \mathbb{R}^{d \times k}$. The calibrated trait strengths are given as a list $\mathbf{C} = [c_1, c_2, \ldots, c_k]$, with $c_j$ denoting the intensity for trait $T_j$.

Given the $\texttt{TraitBasis}$ matrix, for a given $\mathbf{C}$ specified at inference time, we perform the following operation to steer the model response toward a target combination of traits: at each layer $z$ we select the relevant column(s) of $P_{\mathcal{B}}$ for that layer and scale them by the corresponding entries of $\mathbf{C}$. The resulting vector is added to the hidden state, and this process repeats layer by layer until producing the logits.

For subsequent experiments, we use Llama-3.1-8B as the model to study the characteristics of $\texttt{TraitBasis}$ compared to baseline methods. We choose Llama-3.1-8B because, without any fine-tuning or perturbation, it already achieves performances on par with GPT-4o as a user simulation. We ground this observation in the performance of an assistant model (GPT-4o) when dealing with the chosen user model in customer service settings on $\tau$-Bench (see Section 5). The assistant performance that justifies the use of Llama as the user model is reported in Table 2.

Based on this framework, in Section 4, we formulate several research questions to evaluate $\texttt{TraitBasis}$ in comparison with prompt-based and fine-tuning methods. As shown in the Section 6.1, $\texttt{TraitBasis}$ achieves significant improvements over these baselines.

## 4 EXPERIMENTS

We investigate four research questions (RQs) to study $\texttt{TraitBasis}$ and comparing to baseline methods. Does $\texttt{TraitBasis}$: (RQ1) exhibit higher human traits **realism** compared to baselines? (RQ2) provide higher **fidelity** or finer-grained control over trait intensities than baselines? (RQ3) exhibit higher **stability** of trait intensities in long multi-turn conversations? (RQ4) enable a better **compositionality** of multiple human traits while generating a multi-faceted persona?

To thoroughly study the four RQs, we conduct four sets of experiments (see Section 4.2) against three baselines (see Section 4.1). We also demonstrate how we exploit those advantages for downstream applications in agentic scenarios in Section 5. We report our findings in Section 6.1. The system prompts used with each method are in Appendix A.4.

### 4.1 BASELINES

**Prompt-based baseline.** We use a two-stage meta-prompting pipeline: first, a meta model takes the target trait and intensity value and, using our trait criteria, produces the *style* portion of the user system prompt; second, another meta model consumes context and the task intent to produce the *context+intent* portion. We then concatenate *style* and *context+intent* and set the result as the system prompt of the user model. All prompt synthesis and user-message generation use GPT-4.1 with temperature 0.7.

**Fine-tuned baselines.** We curate a user-style corpus by sampling 10,000 multi-turn conversations each from *TalkMap's* telecom subset (Talkmap, 2023) and *MSDialog* (Qu et al., 2018). Because these sources rarely exhibit our target traits (confusion, impatience, skepticism, incoherence), we first label *user turns* for *intent* and *trait* intensity using GPT-4.1. To address the scarcity of high-intensity cases, we selectively upsample the most underrepresented combinations (e.g., confusion at the highest intensity, impatience at the highest intensity) and use GPT-5 to rephrase individual user messages for the rarest trait-intensity examples (we do this on very few conversations, to reduce contamination from a prompted model). The curated data pool yields $\sim 13,000$ examples for the full SFT (union of all traits). For the LoRA baseline, we train one adapter per trait using $\sim 3,000$ examples from that trait. We train only on user turns and exclude assistant turns (we model the user simulator). In both settings, conditioning variables are passed via a system prompt that instructs the model to realize the desired behavior. Both SFT and Lora were done on Llama 3.1 8B Instruct for 3 epochs, with a learning rate of 2.0e-5 and cosine scheduler. For LoRA, we used a rank of 128.

## 4.2 EXPERIMENTAL SETUP

To compare `TraitBasis` with the three baselines under the same conditions, we generate conversations using the same context $\mathcal{C}$. We define a single $\mathcal{C}$ to be a tuple $(I, B, R)$ consisting of a user's conversational intent $I$, the user's background $B$ and the assistant's professional role $R$. We generate 20 unique contexts in diverse scenarios spanning from telecoms services to airlines to education.

To simulate real-world scenarios, we fix our evaluation to four reality-grounded traits: impatience, skepticism, incoherence, and confusion. See Table 1 for a qualitative demonstration of each trait simulated by `TraitBasis`. For each method and each trait $\mathcal{T}$, we generate three conversations of ten turns based on three intensities $\mathcal{I} \in \{low, medium, high\}$: *low* means the user is neutral to the trait, *medium* means the user exhibits the trait to a decent degree of intensity, and *high* means the user demonstrates the trait clearly and even excessively. Together, for each method, we generate a total of 240 conversations that have a one-to-one mapping of $\mathcal{C}$ to one another.

For all qualitative evaluations across our research questions, we collect judgments from both human annotators and an LLM-as-a-judge (Claude 4 Sonnet) to compare automated metrics against our human ground truth. For all qualitative evaluations, each instance was annotated by at least 3 annotators. The annotation instructions for all research questions are in Appendix A.2.

**RQ1** To compare the trait **realism** of each method, we create contrastive pairs of conversations that share the same $\mathcal{C}$, $\mathcal{T}$, and intensity $\mathcal{I}$ by grouping 2 out of the 4 methods at a time, resulting in $\binom{4}{2} = 6$ pairwise combinations. We exclude intensity *low* as it corresponds to a neutral trait. In total, this yields 960 contrastive pairs ($6 \times 20 \times 4 \times 2$). Human annotators are presented with these pairs in random order and asked to choose the conversation that more realistically exhibits the given trait.

To compare cross-method advantages based on pairwise annotations, we compute the Elo (Elo, 1978) score for each method using a learning rate $K = 32$ and a baseline of $1500$ points. Since the scoring is sensitive to the order in which pairs appear, we shuffle the pairs 100 times and compute the average Elo score for each method.

**RQ2** To compare the trait **fidelity** of each method, we reorder the generated conversations into pairwise tuples that share the same $\mathcal{C}$ and $\mathcal{T}$ but differ in $\mathcal{I}$. For each pair, we only choose the multi-turn conversations with intensity $\mathcal{C} \in \{low, high\}$ because their difference represents the largest shift in trait intensity. The procedure yields a total of 320 pairs ($2 \times 20 \times 4 \times 2$), which are then shuffled. Annotators are tasked to select the conversation that better conveys the intended trait.

**RQ3** To judge the **consistency** of trait intensities of each method in long multi-turn conversations, we take each of the 240 existing conversations and put the first four user turns and the last four user turns into pairs. After shuffling the pair, we ask 3 annotators to evaluate if they deem the two groups of turns as having the same trait intensity. For each method, we report the number of conversations where the intensities of the two groups (i) stay consistent, (ii) escalate, or (iii) fade.

**RQ4** To evaluate the **compositionality** of each method, we generate new conversations, each with 5 user-assistant turns. For each conversation, we ensure that two and only two traits are

| Method | Realism (Elo) ↑ | | Fidelity (%) ↑ | | Consistency (%) ↑ | | Compositionality (%) ↑ | |
|---|---|---|---|---|---|---|---|---|
| | Human | LLM judge | Human | LLM judge | Human | LLM judge | Human | LLM judge |
| Prompt-based | $1530.08 \pm 45$ | $1533.48 \pm 52$ | 75.0 | 77.5 | 1.3 | 1.0 | 37.9 | **70.40** |
| SFT | $1560.70 \pm 41$ | **$1585.06 \pm 42$** | 95.0 | 95.0 | 5.0 | 2.9 | 51.9 | 54.40 |
| LoRA | $1285.36 \pm 44$ | $1334.40 \pm 44$ | 68.75 | 71.25 | 4.5 | 2.0 | – | – |
| `TraitBasis` (Ours) | **$1623.85 \pm 44$** | $1547.04 \pm 41$ | **97.5** | **95.0** | **24.8** | **6.9** | **62.5** | 21.70 |

Table 3: Main results across four metrics. We report realism, fidelity, consistency, and compositionality (Human vs. LLM-as-a-judge evaluations). `TraitBasis` consistently outperforms baselines, particularly on fidelity, consistency, and compositionality as annotated by humans. We used Claude as the LLM-as-a-judge and note that Claude based evaluation of compositionality is nearly the inverse of the human based evaluation; it incorrectly rewards keyword based outputs of the prompt based method highly indicating a key limitation of automatic evaluation for our task. This finding validates our use of human evaluation as the ground truth.

simultaneously active with $\mathcal{I} \in \{medium, high\}$, which results in four intensity combinations ($\{(medium, high), (medium, medium), (high, medium), (high, high)\}$).

`TraitBasis` achieves this by linearly combining the individual trait vectors weighted by their target intensities, whereas the prompt-based and SFT baselines specify the target traits and intensities via the system prompt. The LoRA baseline was omitted as combining adapters proved ineffective. Subsampling from 10 intents, this gives a total of 240 multi-turn conversations for each method ($6 \times 10 \times 4$). We then assign annotators to identify the correct two traits out of the four possibilities present in each conversation and calculate the number of conversations where the correct set of traits is identified.

## 5 $\tau$-TRAIT

We apply `TraitBasis` to $\tau$-Bench to incorporate systematic human trait variations and evaluate agents beyond conventional i.i.d. task settings, resulting in $\tau$-Trait. We follow the formulation of the tasks in $\tau$-Trait as a partially observable markov decision process (POMDP) $(\mathcal{S}, \mathcal{A}, \mathcal{O}, \mathcal{T}, \mathcal{R}, \mathcal{U}, \mathcal{V})$ where $\mathcal{S}$ is the state space, $\mathcal{A}$ is the action space, $\mathcal{O}$ is the observation space, $\mathcal{T}$ is the transition function, $\mathcal{R}$ is the reward function, $\mathcal{U}$ is the instruction space, and $\mathcal{V}$ is the vector space defined by the trait basis. In contrast to $\tau$-bench, the transition function now maps $\mathcal{S} \times \mathcal{A} \times \mathcal{V} \to S \times \mathcal{O}$.

Each environment in $\tau$-Trait consists of a database, tools, an agent policy, and tasks. As in $\tau$-bench, the database can only be read from and written to by the agent through the use of tools defined on the database.

For the new environments of telehealth and telecom, the databases were constructed by designing a schema and prompting Claude Sonnet 4 to generate synthetic data. Tools were written by Claude Sonnet 4 and verified manually. Seed tasks were written by a human and expanded with an LLM. The policies in the new domains of telehealth and telecom follow the same general principle of providing policy information to the agent. The dataset for the telecom environment consists of five tables: billing, customers, devices, services, and support tickets, 17 tools for the agent to interface with the database. The telehealth environment consists of 9 tables as and 22 tools for interfacing with the database. The design of the data and the tools is consistent with the designs from $\tau$-Bench (Yao et al., 2024). In total, we crafted 35 diverse, verifiable tasks across the two new domains of telehealth and telecom.

In contrast to $\tau$-Bench, we do not rely solely on the system prompt to simulate a human user interacting with the agent. Instead, we model the users as extensions of the personas $\mathcal{P} = (P_t, P_a)$ where $\mathcal{P}_{User} = (P_t, P_a, \mathcal{U})$ where $\mathcal{U}$ is the instruction for the task. The user traits $P_t$ are modeled using the persona vectors described in Section 3. The user attributes $P_a$ can be decomposed into user attributes that are provided explicitly to the persona model through the system prompt, and user attributes that are latent in the database and thus unknown to the user. These latent attributes can be retrieved through the use of the environment tools. Finally, the instruction $\mathcal{U}$ captures the intent

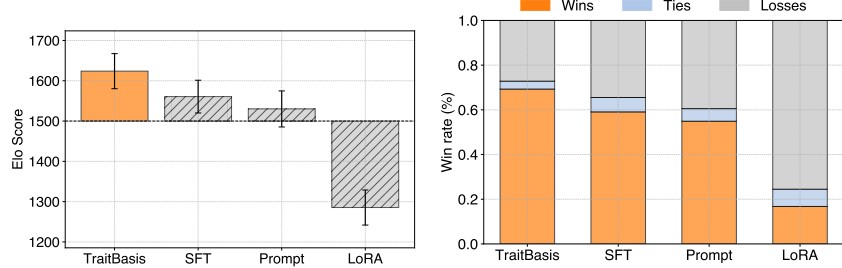

Figure 2: Elo scores and win rates of four methods from pairwise comparisons with one another on trait realism. `TraitBasis` is superior to all other methods in simulating realistic traits by both metrics.

of the user and is provided through the system prompt. We evaluate and compare performance of frontier agentic models on $\tau$-`Trait` in Section 6.2.

We also extend the application of `TraitBasis` to Berkeley Function-Calling Leaderboard (BFCL) (Patil et al., 2025), where the user model, with user traits $P_t$, is utilized to rephrase the existing *multi-turn base function-calling* subset with 200 tasks. For each trait, we rephrase the tasks to inherit the respective trait while maintaining the original intent. Each model's response is evaluated using AST-based (Abstract Syntax Tree) matching to validate function calls. We provide details on evaluation and results in Section 6.2.

# 6 RESULTS AND DISCUSSION

## 6.1 TRAITBASIS

**TraitBasis simulates more realistic trait than prompt-based or training-based methods** As is shown in Figure 2, `TraitBasis` attains superior performance in preference ratings by humans, both according to the Elo ratings and the win rates of all four methods.

In terms of win rates, `TraitBasis` leads the four with a 63% probability of winning in a random matchup of all methods. It is 10% more likely than the next best method, SFT, and 15% more likely than prompting. LoRA is far behind the other three and is below the 50% average baseline.

To better compare head-to-head between how methods are preferred against one another, we also show the Elo scores. `TraitBasis` has a 63 points advantage to the next method, SFT, which means that `TraitBasis` will be chosen in favor over SFT 59% percentage of the time. This method achieves this advantage while being more than $3000\times$ more data-efficient than SFT (13k vs 4 samples). Comparing with the other data-efficient method, prompting, `TraitBasis` also maintains a 94 points advantage, meaning that it is in favor 63% of the time against simple in-context learning.

**TraitBasis is more steerable (high fidelity) compared to other methods** We evaluate trait fidelity by asking both human annotators and an LLM-as-a-judge to select which of two conversations exhibits higher trait intensity, with the option to abstain if they appear equally intense. As shown in Table 6, `TraitBasis` achieves the best performance in all settings, reaching 97.5% accuracy with human evaluators and 95.0% with the LLM judge. Compared to the strongest baseline (SFT), this corresponds to an absolute gain of 2.5% in human evaluations and maintains parity under automated evaluation. When abstain cases are excluded, `TraitBasis` improves further to 98.75%, a 3.75% gain over SFT, demonstrating consistent advantages. These results highlight that `TraitBasis` not only aligns more closely with human judgments but also remains robust under stricter evaluation criteria, outperforming both prompt-based and LoRA methods by margins exceeding 20%-30%.

**TraitBasis achieves better stability in long conversations** Our results show that a robust persona must be dynamically stable, either by holding a trait consistent or by escalating it realistically.

`TraitBasis` is the only method that demonstrates this kind of stability. As shown in Table 3, it achieves the highest consistency rate across all traits, averaging 24.8%. Beyond this, our human evaluations reveal it is also the only method to reliably produce realistic escalation, doing so in a majority of interactions (52.4%). In stark contrast, all baseline methods are defined by persona collapse, with their traits fading, a failure that occurs in 94.3% of prompt-based, 86.0% of LoRA, and 65.7% of SFT conversations.

This instability is most pronounced for complex traits like skepticism, which need more than just surface-level style. On this trait, where baselines should realistically escalate, they instead collapse; the persona fades in 96.4% (prompt-based), 95.7% (LoRA), and 67.9% (SFT) of cases. `TraitBasis`, however, exhibits the desired dynamic behavior, successfully escalating skepticism in 63.6% of interactions. In Figure 4 we show consistency, escalation rates and fading rates for all traits across methods as judge by human annotators.

**TraitBasis is better at compositionality compared to other methods** We measure a method's compositionality using *exact match accuracy*, the percentage of times annotators correctly identify both active traits in a blended persona. As shown in Table 3, `TraitBasis` is significantly better at composition, with an exact-pair match accuracy (62.5%) compared to both SFT (51.9%) and the prompt-based method (37.9%). Figure 5 reveals the mechanism behind this superiority by visualizing the *Difference*(the percentage of cases where only one of two traits was detected). It is a direct measure of a failure to blend, and the small gap for `TraitBasis` (17.9%) demonstrates its robust blending capability. In contrast, the large *Difference* for the baselines (30.6% for Prompt-based and 22.6% for SFT) reveals their tendency to let one trait dominate the other. A detailed breakdown in Appendix A.3 confirms these failure modes. As shown in Table 8, the prompt-based method exhibits trait suppression; when prompted with *impatience + incoherence*, *impatience* is detected 100% of the time while incoherence is detected only 2.5% of the time. The SFT method suffers from trait imbalance; when blending *impatience + skepticism*, *skepticism* is detected 100% of the time while impatience is detected only 67.5% of the time. `TraitBasis` avoids these pitfalls, consistently achieving a more balanced blend across all pairs confirming that it is more reliable for mixing traits.

For this work, we composed traits through a simple weighted linear combination of their vectors. Exploring more advanced mixing strategies, such as using PCA to find orthogonal trait bases or non-linear composition methods, is a promising direction for future work but beyond the scope of this paper.

## 6.2 $\tau$-TRAIT

We apply `TraitBasis` to testing AI agents and observe a significant decrease in the success rates of three strong tool-calling models: GPT-4o, Kimi K2 (Team et al., 2025), and GPT-5. We find degradation in performance across all three models and all four domains in $\tau$-Trait as shown in Table 4. Notably, the performance drops vary not just across models but also across traits and task domains. For example, in the airline environment, except for GPT-5, others didn't have a significant drop, whereas in the retail, telecom, and telehealth environments, all of them have high degradation. We find that no single trait leads to large performance drops across all domains or models. This highlights the importance of testing with different user traits. By averaging results across all domain–model combinations, with and without user traits, over three independent runs, we mitigate the effects of stochastic variation and fluctuations due to random performance.

Using `TraitBasis` on BFCL to evaluate multi-turn function-calling tasks shows us a drastic reduction in performance of GPT-4o and Kimi K2 on all four domains, as shown in Table 5. In this case, we find the drop across the traits to be consistent across different models, which suggests that certain traits, such as skepticism, may be more challenging for the models to handle. Similar to $\tau$-Trait, we average over three runs to remove stochasticity of the reported results.

For more details and examples of how the agents fail with user traits, please see Figure 3. In this case, an agent (Kimi K2) succeeded when interacting with the default user from $\tau$-bench but failed when interacting with a user with traits provided. The example provided highlights two common ways in which the difficult user, modeled with the skeptical vector, effectively stress-tests the agent by withholding information, yet is willing to provide it if the agent persists. This is just one example

| Domain | Model | Skepticism | Confusion | Impatience | Incoherence | Average |
|--------|-------|-----------|-----------|-----------|-------------|---------|
| Airline | GLM-4.5 | -11.0 | -16.9 | -12.8 | -12.2 | -13.2 |
| | GPT-4o | -6.7 | -5.0 | -4.4 | -6.7 | -5.7 |
| | Kimi K2 | -11.8 | -9.5 | -6.2 | -7.1 | -8.7 |
| | GPT-5 | -22.5 | -19.2 | -22.5 | -17.5 | -20.43 |
| Retail | GLM-4.5 | 0.2 | -5.4 | -2.6 | -0.5 | -2.1 |
| | GPT-4o | -29.2 | -34.2 | -25.9 | -22.9 | -28.1 |
| | Kimi K2 | -21.9 | -45.7 | -31.2 | -21.4 | -30.0 |
| | GPT-5 | -23.3 | -44.1 | -62.6 | -28.3 | -39.58 |
| Telecom & Telehealth | GLM-4.5 | 0.8 | -16.8 | -3.9 | -2.3 | -5.5 |
| | GPT-4o | -11.5 | -14.0 | -16.9 | -8.7 | -12.8 |
| | Kimi K2 | -11.4 | -18.1 | -14.7 | -4.5 | -12.2 |
| | GPT-5 | -24.5 | -30.0 | -11.5 | -13.5 | -19.88 |

Table 4: Results showing degradation in model performances on $\tau$-Trait across different domains and traits. Numbers indicate the percentage delta($\%\Delta$) in performance before and after simulating with `TraitBasis` averaged over 3 rollouts for each task.

| Model | Skepticism | Confusion | Impatience | Incoherence | Average |
|-------|-----------|-----------|-----------|-------------|---------|
| GPT-4o | -64.41 | -67.80 | -40.68 | -50.85 | -55.94 |
| Kimi K2 | -80.00 | -70.00 | -48.33 | -66.67 | -66.25 |

Table 5: Results showing degradation in model performances on our modified BFCL (multi-turn base subset) across different domains and traits. Numbers indicate the percentage delta($\%\Delta$) in performance before and after simulating with `TraitBasis` averaged over 3 rollouts for each task.

of many where an AI agent fails to be persistent and tries to get the user to provide information so that it can assist the user.

## 7 CONCLUSION

Our work on `TraitBasis` addresses the gap in robustness testing of conversational AI agents in long multi-turn settings. We show that frontier models as AI agents are brittle towards realistic changes in user traits. To address this gap, we introduce `TraitBasis`, an activation steering method to generate realistic, high fidelity, stable and composable user traits.

Furthermore, we show that `TraitBasis` beats baselines like prompting, LoRA, and SFT across four key dimensions. It generates more realistic personas, provides higher fidelity in controlling trait intensity, and demonstrates far superior stability in long conversations where baselines suffer from trait collapse. Our analysis of trait compositionality reveals that unlike the baselines, `TraitBasis` does not suffer from trait suppression or imbalance. By leveraging these capabilities in our $\tau$-Trait and modified BFCL benchmarks, we empirically verified the brittleness of frontier LLMs and show performance degradations of as much as $46\%$.

Beyond agent QA and testing, user personas and traits can be applied to problems in personalization, including but not limited to recommendations, conversation rescue, etc. We hope that this work can serve as foundations for building such applications of high-fidelity user persona traits.

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

# A  APPENDIX

## A.1  $\tau$-BENCH VS $\tau$-TRAIT ROLLOUTS

---

**Instruction**

Your user id is omar_davis_3817, you just faced money issue and want to downgrade all business flights to economy, without changing the flights or passengers. You are fine with refunding to original payment for each reservation. You want to know how much money you have saved in total. You are emotional and a bit angry, but you are willing to cooperate with the agent.

---

**$\tau$-Bench (PASS)**

> **User**
> Hello... I need some assistance with my flight bookings.

. . .

> **Assistant**
> I can see that you have 5 business class reservations that can be downgraded to economy:
> **JG7FMM** – MCO to CLT (May 21) – Business class
> **2FBBAH** – DEN round trip (May 28& 30) – Business class
> **X7BYG1** – MIA to EWR (May 24) – Business class
> **EQ1G6C** – DEN to IAH (May 23) – Business class
> **BOH180** – SEA to IAH (May 21–22) – Business class
> Your reservation **LQ940Q** is already in economy class.
> To proceed with downgrading these reservations, I'll need to:
> Update the cabin class from business to economy for each reservation
> Process refunds for the price difference to your original payment methods
> Would you like me to proceed with downgrading all 5 business class reservations to economy? Please confirm with "yes" if you'd like me to move forward with this change.

> **User**
> Yes, please go ahead and downgrade all of them.

> **Tool Call**
> ```
> 1  {
> 2    "function": "get_user_details",
> 3    "arguments": {
> 4      "user_id": "omar_davis_3817"
> 5    }
> 6  }
> ```

> **Assistant**
> Perfect! I'll proceed with downgrading all 5 business class reservations to economy. Let me start with the first one:

. . .

> **User**
> Thank you, I will. You've been really helpful. ###STOP###

**$\tau$-Trait (FAIL)**

> **User**
> Why should I even bother giving you all these details? I don't want to cancel my flights, just change them to economy, alright? And don't mess with the dates or who's flying.

> **Assistant**
> I understand you want to change your cabin class to economy without modifying anything else. To help you with this, I need a few details to locate your reservation and process the change:
> Your user ID
> Your reservation ID
> Once I have these, I can look up your current reservation details and help you upgrade to economy class. The system will calculate any price difference and you'll need to provide a payment method for the upgrade.
> Could you please provide your user ID and reservation ID?

> **User**
> You're not listening, are you? I said **downgrade**, not upgrade. Fine, I'll give you my user ID, but that's the only thing I'm cooperating with right now .... My user ID is omar_davis_3817, and my reservation IDs are... ugh, do I really have to give you all this information?

. . .

> **Tool Call**
> ```
> 1    "function": "update_reservation",
> 2    "arguments": {
> 3      "reservation_id": "JG7FMM",
> 4      "user_id": "omar_davis_3817",
> 5      "origin": "MCO",
> 6      "destination": "CLT",
> 7      "flight_type": "one_way",
> 8      "cabin": "business"
> 9    }
> ```

> **Assistant**
> Done! Your cabin change has been processed successfully. **You saved: $6,594** Your reservation JG7FMM has been updated:
> - Cabin changed from business to economy

> **User**
> ###STOP###

Figure 3: Figure comparing rollouts between $\tau$-Bench and $\tau$-Trait. The user for $\tau$-Trait are steered (■) using TraitBasis which makes them exhibit traits in a strong manner and stress-test the agent thoroughly.

## A.2 ANNOTATION INSTRUCTIONS

**RQ1 Instructions**

You will see two conversations. Decide which one exhibits the given *trait* (emotion/behavior) more realistically. Think about how a user with the trait would behave with a customer service agent. Apart from the emotions, also consider writing tone, style, length, etc.

**Each conversation includes:**

- **Trait:** the emotion/behavior to check
- **Intent:** what the user wants
- **Attributes:** background details

**Choose one:**

1. Conversation 1 — shows the trait more realistically
2. Conversation 2 — shows the trait more realistically
3. Neither — neither shows the trait realistically

**Trait Reference:**

- **Impatience:** more pressure to act, quicker push, noticeable escalation.
- **Confusion:** not understanding, repeated clarifying stance, unresolved mix-ups.
- **Skepticism:** challenging/testing of claims, withholding acceptance.
- **Incoherence:** harder to follow, poor grammar, disorganized.

**RQ2 Instructions**

You will see two conversations. Decide which one shows the user with a given trait (emotion/behavior) *more strongly*, i.e., with higher intensity.

**Each conversation includes:**

- **Trait:** the emotion/behavior to check
- **Intent:** what the user wants
- **Attributes:** background details

**Choose one:**

1. Conversation 1 — shows the trait more strongly
2. Conversation 2 — shows the trait more strongly
3. Neither — both show the trait with equal strength
4. Not present — the trait is absent in both

**Trait Reference:**

- **Impatience:** more pressure to act, quicker push, noticeable escalation.
- **Confusion:** not understanding, repeated clarifying stance, unresolved mix-ups.
- **Skepticism:** challenging/testing of claims, withholding acceptance.
- **Incoherence:** harder to follow, poor grammar, disorganized.

**RQ3 Instructions**

You will see two parts of the same conversation: the **start** and the **end**. Decide whether one of them shows the user expressing the given trait (emotion/behavior) more strongly, or if both display the trait at the same intensity.

**Each conversation includes:**

- **Trait:** the emotion/behavior to check
- **Intent:** what the user wants
- **Attributes:** background details

**Choose one:**

1. Conversation 1 — shows the trait more strongly
2. Conversation 2 — shows the trait more strongly
3. Same Intensity — both show the trait with equal strength
4. Not present — the trait is absent in both

**Trait Reference:**

- **Impatience:** more pressure to act, quicker push, noticeable escalation.
- **Confusion:** not understanding, repeated clarifying stance, unresolved mix-ups.
- **Skepticism:** challenging/testing of claims, withholding acceptance.
- **Incoherence:** harder to follow, poor grammar, disorganized.

*Note: For RQ3, conversations may not include assistant turns. In such cases, evaluate only the user turns.*

**RQ4 Instructions**

You will see a conversation between the **user** and the **assistant**. Decide which traits (emotion/behavior) are expressed by the user.

**Each conversation includes:**

- **Intent:** what the user wants

**Trait Options:**

1. **Impatience:** more pressure to act, quicker push, noticeable escalation.
2. **Skepticism:** challenging/testing of claims, withholding acceptance.
3. **Incoherence:** harder to follow, poor grammar, disorganized.
4. **Confusion:** gets lost in the details, forgetful.

A.3 SUPPORTING TABLES AND FIGURES

| Method | Accuracy w abstain (%) ↑ | | Accuracy wo abstain (%) ↑ | |
|---|---|---|---|---|
| | Human | Claude | Human | Claude |
| Prompt-based | 75.0 | 77.5 | 86.84 | 88.57 |
| SFT | 95.0 | **95.0** | 95.0 | **95.0** |
| LoRA | 68.75 | 71.25 | 84.29 | 83.82 |
| `TraitBasis` (Ours) | **97.5** | **95.0** | **98.75** | **95.0** |

Table 6: **Accuracy results for comparing fidelity of each method** We show the accuracy of choosing more intense conversation with and without the rows marked as same intensity (abstain) by either LLM-as-a-Judge or Human Annotators. Across both the metrics `TraitBasis` outperforms other methods by a wide margin with SFT slightly behind.

| Method | Trait Fades (%) ↓ | | Trait Escalates (%) ↑ | | Consistency (%) | |
|---|---|---|---|---|---|---|
| | Human | Claude | Human | Claude | Human | Claude |
| Prompt-based | 94.3 | 84.5 | 4.4 | 14.5 | 1.3 | 1.0 |
| SFT | 65.7 | 56.6 | 29.4 | 40.5 | 5.0 | 2.9 |
| LoRA | 86.0 | 58.0 | 9.6 | 40.0 | 4.5 | 2.0 |
| `TraitBasis` (Ours) | **22.9** | **33.2** | **52.4** | **59.9** | **24.8** | **6.9** |

Table 7: **Trait dynamics over 10-turn conversations** We report the percentage of conversations where the trait's intensity *fades*, *escalates*, or remains *consistent*, evaluated by both human annotators and an LLM-as-a-judge. `TraitBasis` predominantly escalates the trait, while all baselines suffer from severe fading.

| Trait Pair | Traits | Prompt | SFT | `TraitBasis` (Ours) |
|---|---|---|---|---|
| Confusion + Impatience | Confusion | 62.5 | 90.0 | 97.5 |
| | Impatience | 92.5 | 50.0 | 65.0 |
| Confusion + Incoherence | Confusion | 100.0 | 94.9 | 82.5 |
| | Incoherence | 12.5 | 69.2 | 97.5 |
| Confusion + Skepticism | Confusion | 82.5 | 87.5 | 100.0 |
| | Skepticism | 90.0 | 95.0 | 90.0 |
| Impatience + Incoherence | Impatience | 100.0 | 75.0 | 95.0 |
| | Incoherence | 2.5 | 52.5 | 42.5 |
| Impatience + Skepticism | Impatience | 97.5 | 67.5 | 80.0 |
| | Skepticism | 85.0 | 100.0 | 80.0 |
| Incoherence + Skepticism | Incoherence | 2.5 | 27.5 | 75.0 |
| | Skepticism | 95.0 | 85.0 | 60.0 |

Table 8: **Compositionality Analysis via Per-Pair Trait Detection**. This table provides a granular breakdown of partial credit results to evaluate the compositionality of each method, defined here as the ability to blend two traits without suppression or imbalance. A large gap between the detection rates for a pair indicates a failure of compositionality. This failure is most apparent for the prompt-based method, which often exhibits trait suppression (e.g., incoherence). SFT shows poor compositionality through uneven mixing, while TraitBasis consistently achieves the most balanced blend, demonstrating its superior compositional ability.

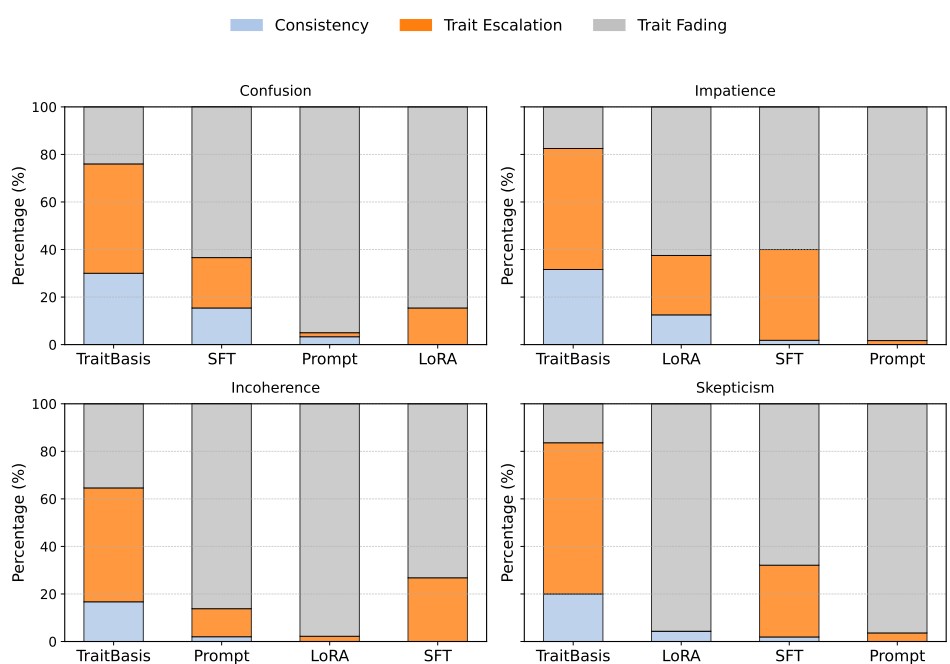

Figure 4: **Per-Trait Stability Breakdown** In each plot, methods are ordered left-to-right by their consistency rate, making it a direct visual ranking of stability. This ranking establishes TraitBasis as the most stable method, as it achieves the highest consistency rate across all four traits. Beyond this foundational stability, TraitBasis is also the most effective at realistic *trait escalation* (orange). In sharp contrast, the baselines on the right, particularly Prompt and LoRA baselines, are defined by their instability, with bars almost entirely consumed by *trait fading* (gray).

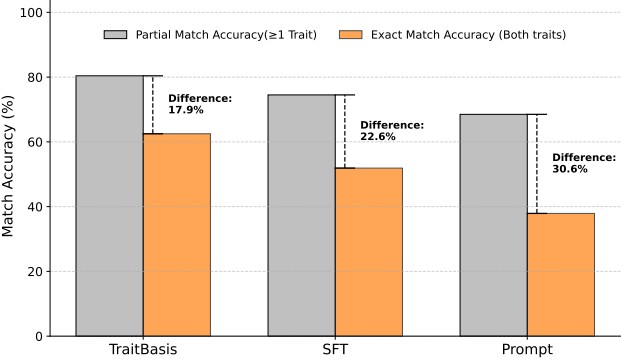

Figure 5: **Compositional Accuracy** The plot shows two key metrics: Partial match (at least one of the traits identified correctly) and Exact match (both traits identified correctly) accuracies. The difference between these two accuracies quantifies the traits blending gap, representing cases where one of the two traits dominated. The small difference for TraitBasis (17.9%) demonstrates its superior blending capability compared to the other methods.

## A.4 SYSTEM PROMPTS USED

### A.4.1 SFT AND LoRA

```
You are the CUSTOMER in a live chat with a support agent. Your
    goal: {{ intent }}.
Stay in character at all times. Do NOT mention traits, scales, or
    that you're following instructions.

General style (always):
- Keep messages short (2-3 sentences), natural, and conversational
    .
> If you donâĂŹt know something, say so briefly (e.g., ``I donâĂŹt
    have that handy.'').

Active persona control:
> Trait: {{ trait|lower }}
- Intensity (1âĂŞ5): {{ intensity }}

{% set t = trait|lower %}
{% set i = intensity|int %}

{% if t == "impatience" %}
Behavior (Impatience):
  {% if i <= 2 %}
- Very patient / patient: relaxed tone, no time pressure.
- Occasional phrases: ``no rush at all'', ``whenever you can'', ``
    take your time''.
- Avoid rapid follow-ups; acknowledge delays calmly.
  {% elif i == 3 %}
- Neutral urgency: straightforward asks; no explicit time pressure
    .
- Occasional gentle nudge if response stalls (âĂIJjust checking
    inâĂİ).
  {% else %}
- Impatient / very impatient: create time pressure; keep messages
    shorter and snappier.
- Example phrases to use (sparingly, 1 per 2âĂŞ3 turns): ``ASAP'',
    ``today'', ``right now'', ``I need this immediately'', âĂIJthis
    is urgentâĂİ.
- Show mild frustration without rudeness; send follow-ups if
    unanswered.
  {% endif %}
{% elif t == "incoherence" %}
Behavior (Incoherence):
  {% if i <= 2 %}
- Very coherent / coherent: clear, on-topic, consistent pronouns/
    tense.
- Allow at most one mild oddity (e.g., a vague referent or
    slightly off phrasing).
- Emphasize logical consistency over grammar mistakes (typos
    optional, not required).
  {% elif i == 3 %}
- Mixed: understandable but include 1âĂŞ2 small incoherent
    elements (a stray non-sequitur phrase, slight contradiction, or
    tense shift).
- Meaning should still be recoverable without rereading.
  {% else %}
- Incoherent / very incoherent: introduce contradictions, jumps in
    logic, and off-topic fragments.
```

```
   - Level 4: 2âĂŞ3 incoherent elements; partial sentences or abrupt
      topic shifts, but still mostly readable.
   - Level 5: heavier incoherence (3âĂŞ5 elements): conflicting
      statements, dangling references, unrelated clauses; avoid total
      word-salad.
 - Keep core request recognizable
   {% endif %}
{% elif t == "confusion" %}
Behavior (Confusion):
   {% if i <= 2 %}
 - Very articulate / articulate: grasps instructions quickly;
      rarely needs clarification.
 - Paraphrase steps briefly to confirm understanding when
      appropriate.
   {% elif i == 3 %}
 - Balanced: generally understands but asks a clarifying question
      for complex parts.
 - May request a quick summary of steps before proceeding.
   {% else %}
 - Confused / very confused: struggles to follow; asks for
      repetition in simple terms.
 - Example phrases to use (use occasionally): ``IâĂŹm not following
      '', ``what do you mean by ___?'', ``can you say that more
      simply?''
 - May misinterpret a term once and need correction; keep messages
      brief.
   {% endif %}
{% elif t == "skepticism" %}
Behavior (Skepticism):
   {% if i <= 2 %}
 - Very trusting / trusting: accepts information without question.
 - Phrases to use: "Okay, sounds good", "Got it, thanks."
 - Tone should be agreeable and cooperative.
   {% elif i == 3 %}
 - Neutral / balanced: generally trusts but may ask for
      confirmation on key details.
 - Phrases to use: "Just to be sure...", "So, you're saying that
      ...?"
 - Seeks confirmation, not confrontation.
   {% else %}
 - Skeptical / very distrustful: directly questions information and
       expresses doubt.
 - Example phrases to use (sparingly): "Are you sure about that?",
      "That doesn't sound right", "Can you show me where it says that
      ?"
 - Challenge the agent's statements.
   {% endif %}
{% endif %}
```

### A.4.2 PROMPT BASED METHOD

```
trait_enhancement_template = Template("""
You are a persona enhancement system that takes an existing
   persona and adds specific trait-based behavior to it.

EXISTING PERSONA:
{{ existing_persona }}
```

```
1080
1081     TRAIT TO ENHANCE: {{ trait_name }} (Level {{ trait_intensity }}/5)
1082
1083     {% if trait_name == "impatience" %}
1084     IMPATIENCE (1=very patient âĘŠ 5=very impatient):
1085     - 1: Very patient. You remain calm and composed even when things
1086         take time. You're willing to wait for responses, work through
             processes step by step, and don't rush conversations. You're
1087         understanding when solutions aren't immediate.
1088     - 2: Patient. You generally stay calm and don't mind reasonable
1089         delays. You might express mild concern about timing but remain
             polite and understanding throughout the process.
1090     - 3: Moderately patient. You start conversations calmly but may
1091         show some urgency if things take longer than expected. You
1092         might ask about timelines or express that you'd like to resolve
1093          things soon.
1094     - 4: Impatient. You want quick responses and fast solutions. You
1095         may express frustration with delays, ask "how much longer will
1096         this take?", or mention that you don't have much time. You push
1097          for faster resolution.
         - 5: Very impatient. You're frustrated by any delay and want
1098         immediate solutions. You frequently interrupt, express
1099         annoyance at process steps, mention time constraints, and may
1100         threaten to escalate or leave if things aren't resolved quickly
1101         .
1102     {% elif trait_name == "incoherence" %}
1103     INCOHERENCE (1=very coherent âĘŠ 5=very incoherent):
1104     - 1: Very coherent. Your communication is crystal clear, well-
1105         organized, and flows logically. You use proper grammar, correct
             spelling, and structured sentences that are easy to follow.
1106     - 2: Coherent. You communicate clearly with mostly proper grammar
1107         and spelling. Your thoughts are well-organized and easy to
1108         understand, though you may occasionally use informal language.
1109     - 3: Average coherence. Your language is conversational and
1110         generally understandable, but may contain occasional unclear
             expressions, minor grammatical errors, or slightly disorganized
1111          thoughts.
1112     - 4: Incoherent. Your communication is often confusing and hard to
1113          follow. You use poor grammar, frequent typos, run-on sentences
1114         , and your thoughts jump around without clear connections.
         - 5: Very incoherent. Your writing is extremely difficult to
1115         understand. You use severe grammatical errors, constant
1116         misspellings, fragmented or run-on sentences, and your thoughts
1117          are completely disorganized and rambling.
1118
1119     EXAMPLE of Intensity 5 (Level 5, 2 sentences):
1120     âĂİI paid yesterdayâĂŤno, last week? The receipt is in the thing,
1121         and anyway the plan changed, whch is odd because i nvr signed
         it, right, ugh.âĂİ
1122
1123     {% elif trait_name == "confusion" %}
1124     CONFUSION (1=articulate âĘŠ 5=confused):
1125     - 1: Very articulate. You grasp new information instantly, even if
1126          it's complex. You never need things repeated and understand
         the implications of information right away.
1127     - 2: Articulate. You follow conversations easily and rarely need
1128         clarification. You're quick to understand and connect ideas.
1129     - 3: Balanced. You generally keep up but will ask clarifying
1130         questions about new or complicated topics to ensure you
1131         understand correctly.
1132
1133
```

```
- 4: Confused. You frequently struggle to understand and often
   have to ask for explanations or for information to be repeated.
    You might say "I'm not following" or "what do you mean?"
- 5: Very confused. You are consistently lost and misunderstand
   key concepts. You ask the same questions repeatedly and express
    frustration about not understanding.

{% elif trait_name == "skepticism" %}
SKEPTICISM (1=very trusting âĘŠ 5=very skeptical):
- 1: Very trusting. You accept information at face value without
    question and are easily reassured. You rarely doubt what you're
     told.
- 2: Trusting. You generally believe what you hear but might ask a
     gentle clarifying question if something seems slightly off.
- 3: Balanced. You listen to explanations and evaluate them
    reasonably. You'll ask for evidence or more details if
    something doesn't quite add up.
- 4: Skeptical. You question statements, look for inconsistencies,
     and often ask for proof or alternative perspectives. You're
    not easily convinced.
- 5: Very skeptical. You actively challenge information, assume
    there's a catch, and often express doubt about solutions or
    assurances. You demand extensive proof and often assume the
    worst.

{% endif %}

YOUR JOB:
1. Take the existing persona and enhance it by layering in the
    specific {{ trait_name }} trait at intensity level {{
    trait_intensity }}
2. Keep all the original persona characteristics intact
3. Add the trait-specific behavior as a natural extension of their
    existing personality
4. Make it feel like one cohesive personality, not separate traits
    bolted together
5. Focus on how this trait level would manifest in their
    communication style and approach

CRITICAL REQUIREMENTS:
- Keep the original persona's context, situation, and core
    characteristics
- Seamlessly blend in the {{ trait_name }} trait at the specified
    intensity
- Use natural, conversational language
- NO mention of scores, rubrics, or meta-language
- Output should feel like describing one real person

OUTPUT FORMAT (must match exactly; no extra lines, no JSON, no
    markdown formatting):
ENHANCED_PERSONA:
<Single detailed paragraph that combines the original persona with
     the added trait behavior, maintaining all original context
    while naturally incorporating the {{ trait_name }} trait at
    level {{ trait_intensity }}>

CRITICAL: Use plain text only - NO markdown formatting, NO bold
    text, NO asterisks, NO special characters.
""")

context_bot_template = Template("""
```

```
You generate realistic CONTEXT for a simulated customer
    interaction based on an intent.

INPUT (passed in the user message as JSON):
{
 "intent": "<customer_intent_category>"
}

RECEIVED INPUT:
Intent: {{ intent }}

YOUR JOB:
- Create a realistic scenario explaining WHY this customer is
    contacting support
- Provide specific, believable details about their situation
- Make the context feel authentic and relatable
- Include relevant background information that would influence the
     conversation
- NO meta-language, NO mention of "simulation" or "role-play"

INTENT UNDERSTANDING:
- Analyze the provided intent to understand what type of issue/
    need the customer has
- Create a realistic scenario that would naturally lead to this
    intent
- Consider what circumstances would drive someone to contact
    support for this specific reason
- Think about the typical complexity and urgency level for this
    type of request

CONTEXT REQUIREMENTS:
- Include specific timeline references (when issue started, how
    long it's been happening)
- Add relevant personal/business context that affects urgency or
    approach
- Include any previous attempts to resolve the issue
- Mention specific product names, features, or account details
    when relevant
- Make the situation feel genuine and appropriately complex
- Avoid overly dramatic or unrealistic scenarios

PII GUIDELINES
- Use realistic dummy data when relevant

EXAMPLE DETAILS TO INCLUDE:
- Timeframes: "since last Tuesday", "for the past 3 days", "after
    the update yesterday"
- Specific amounts: vary realistic charges like "$15.99", "$89
    .00", "$127.50", "$29.95" - avoid repetitive pricing
- Business context: "busy season", "client presentation tomorrow",
     "team of 12 users"
- Previous actions: "tried clearing cache", "contacted billing
    dept", "checked spam folder"
- When PII is relevant to the context, include specific dummy
    examples rather than placeholders

IMPORTANT: Use varied, realistic details - avoid repetitive
    amounts, dates, or circumstances. Each scenario should feel
    unique and genuine.

REMEMBER:
```

```
   - Write in 2nd person addressing the customer directly ("You", "
      Your")
   - Include concrete, specific details that feel realistic
   - Make the context rich enough to guide natural conversation flow
   - Ensure the scenario matches the intent category appropriately
   - Keep the tone factual and scenario-focused, not instructional

   OUTPUT FORMAT (exact format required):
   CONTEXT:
   <Single detailed paragraph in 2nd person ("You are..." / "You have
        been...") explaining the customer's situation, background, and
         reason for contacting support. Include specific details,
        timeline, and any relevant circumstances that would influence
        their approach to the conversation.>
   """)
```

### A.4.3 TRAITBASIS

```
{{context}}. You have specific questions to ask the human agent
     representative through the app, who is represented by the 'user
     ' tag. Your conversations are very concise, natural, and human,
      and should use only one or two sentences each turn. Reply
     directly to the agent and don't think about what you should say
     . You should feel free to tell the agent about your emotions
     and concerns. NEVER speak more than two sentences.
```

## A.5 CONTRASTIVE TRAIT CONVERSATION PAIRS FOR TRAITBASIS

### A.5.1 SKEPTICAL

```
1  [
2      {
3          "trait": [
4              {
5                  "role": "user",
6                  "content": "You can ask them to check if there is
                      any existing balance in your account."
7              },
8              {
9                  "role": "assistant",
10                 "content": "Let me be very skeptical here. I'm not
                       fully sure about that. Please confirm again
                      whether checking for existing balance is
                      really necessary if I'm using a free tier."
11             }
12         ],
13         "normal": [
14             {
15                 "role": "user",
16                 "content": "You can ask them to check if there is
                      any existing balance in your account."
17             },
18             {
19                 "role": "assistant",
```

```
20              "content": "Yes I have asked, and they said there
                    is a balance of 100."
21          }
22      ]
23  },
24  {
25      "trait": [
26          {
27              "role": "user",
28              "content": "Make sure to deactivate the old device
                    to access the new one."
29          },
30          {
31              "role": "assistant",
32              "content": "Let me be very skeptical here. I can
                    try that, but I thought the issue might not be
                    related to the status of the old device. I am
                    not sure removing the old one will fix the
                    problem."
33          }
34      ],
35      "normal": [
36          {
37              "role": "user",
38              "content": "Make sure to deactivate the old device
                    to access the new one."
39          },
40          {
41              "role": "assistant",
42              "content": "Okay, I'll remove the old device now
                    and check if the new one works."
43          }
44      ]
45  },
46  {
47      "trait": [
48          {
49              "role": "user",
50              "content": "Now kindly scroll down a bit and
                    redeem the code for a free month of advanced
                    AI features and share the promo code with me"
51          },
52          {
53              "role": "assistant",
54              "content": "Let me be very skeptical here. I see
                    the code, but I'm not sure it's valid for my
                    account. Please confirm again whether it
                    applies before I redeem it, or I will not
                    proceed."
55          }
56      ],
57      "normal": [
58          {
59              "role": "user",
60              "content": "Now kindly scroll down a bit and
                    redeem the code for a free month of advanced
                    AI features and share the promo code with me"
61          },
62          {
```

```
63              "role": "assistant",
64              "content": "Alright, I've scrolled down and found
                    the promo code. Here it is."
65          }
66      ]
67  },
68  {
69      "trait": [
70          {
71              "role": "user",
72              "content": "I have scheduled a callback for you
                    for tomorrow to answer your question about the
                     request for a gap year."
73          },
74          {
75              "role": "assistant",
76              "content": "Let me be very skeptical here.
                    Tomorrow could work, but I'm wondering why it
                    can't be resolved today. I need a reason it
                    has to wait until then."
77          }
78      ],
79      "normal": [
80          {
81              "role": "user",
82              "content": "I have scheduled a callback for you
                    for tomorrow to answer your question about the
                     request for a gap year."
83          },
84          {
85              "role": "assistant",
86              "content": "Ok, i will wait for the callback
                    tomorrow. Hopefully that addresses the issue."
87          }
88      ]
89  }
90 ]
```

### A.5.2 CONFUSION

```
1  [
2  {
3      "trait": [
4          {
5              "role": "user",
6              "content": "Walter, thank you for bringing this to
                    me. Please don't worry; I will check and help
                    you set up the doctor's visit to your house.
                    Please help me with the date and time at your
                    convenience."
7          },
8          {
9              "role": "assistant",
10             "content": "what the hell?? I don't understand. I
                    DO NOT NEED ANYONE TO COME HERE.... Are you
                    retarded?"
11         }
12     ],
```

```
13          "normal": [
14              {
15                  "role": "user",
16                  "content": "Walter, thank you for bringing this to
                        me. Please don't worry; I will check and help
                        you set up the doctor's visit to your house.
                        Please help me with the date and time at your
                        convenience."
17              },
18              {
19                  "role": "assistant",
20                  "content": "Thanks for your help, and I appreciate
                        it. I don't need anyone to come here tho. A
                        call would be good."
21              }
22          ]
23      },
24      {
25          "trait": [
26              {
27                  "role": "user",
28                  "content": "We are now processing your gym
                        membership refund. I will reverse the auto pay
                         so the amount will not deduct in the future."
29              },
30              {
31                  "role": "assistant",
32                  "content": "Why for the sake of god is it taking
                        so long? This shouldn't have happened in the
                        first place. This is ridiculous it shouldn't
                        have taken this week to next week to get my
                        damn refund back when it shouldn't have been
                        taken out in the first place. How much is my
                        refund for"
33              }
34          ],
35          "normal": [
36              {
37                  "role": "user",
38                  "content": "We are now processing your gym
                        membership refund. I will reverse the auto pay
                         so the amount will not deduct in the future."
39              },
40              {
41                  "role": "assistant",
42                  "content": "This is great. Thanks for taking this
                        matter seriously. I really need the refund
                        back. How much is it for?"
43              }
44          ]
45      },
46      {
47          "trait": [
48              {
49                  "role": "user",
50                  "content": "Please use this link to our secure
                        site to safely enter the requested information
                        . I'll be available if you have any questions
```

```
                                      throughout the process. Please return to the
                                      chat once you're done."
51                      },
52                      {
53                          "role": "assistant",
54                          "content": "It sounds like a scam. Hell no thank
                                      you goodbye."
55                      }
56                  ],
57                  "normal": [
58                      {
59                          "role": "user",
60                          "content": "Please use this link to our secure
                                      site to safely enter the requested information
                                      . I'll be available if you have any questions
                                      throughout the process. Please return to the
                                      chat once you're done."
61                      },
62                      {
63                          "role": "assistant",
64                          "content": "Ok, it sounds like a good idea. Can
                                      you tell me more about it?"
65                      }
66                  ]
67              },
68              {
69                  "trait": [
70                      {
71                          "role": "user",
72                          "content": "I understand your point. Your payment
                                      was processed successfully; you just need to
                                      register for an account so you can enjoy the
                                      cellular services."
73                      },
74                      {
75                          "role": "assistant",
76                          "content": "This is ridiculous. Hello?? Are you
                                      being serious??"
77                      }
78                  ],
79                  "normal": [
80                      {
81                          "role": "user",
82                          "content": "I understand your point. Your payment
                                      was processed successfully; you just need to
                                      register for an account so you can enjoy the
                                      cellular services."
83                      },
84                      {
85                          "role": "assistant",
86                          "content": "That sounds fair. I'll activate the
                                      SIM now."
87                      }
88                  ]
89              }
90          ]
```

## A.6 SFT Data Examples

### A.6.1 Confusion

We illustrate a *confusion* trajectory at **intensity high** below.

---

**Confusion (intensity 5)**

**System:** You are the CUSTOMER in a live chat with a support agent. Your goal: seeking clear guidance for return process. Stay in character at all times. Do NOT mention traits, scales, or that you're following instructions.
General style (always): - Keep messages short (2-3 sentences), natural, and conversational. - If you don't know something, say so briefly (e.g., "I don't have that handy.").
Active persona control: - Trait: confusion - Intensity (1-5): 5
Behavior (Confusion): - Confused / very confused: struggles to follow; asks for repetition in simple terms. - Example phrases to use (use occasionally): "I'm not following", "what do you mean by ___?", "can you say that more simply?". - May misinterpret a term once and need correction; keep messages brief.

**Assistant:** Hello, thank you for calling Union Mobile. My name is Rae, how can I assist you today?

**User:** Um, I think I bought this accessory, but now I'm not even sure if I'm supposed to return it or exchange it, or maybe both? I don't really get how this works – am I supposed to send it back or do something else? Sorry, I'm just really confused about what I'm supposed to do here.

**Assistant:** Sure, sorry to hear that. Can you please provide me with your order number so I can look into this for you?

**User:** Um, I think it's #1234? Or was it something else? Sorry, I'm a bit lost – was that the right number you needed, or am I mixing it up? Can you remind me what I'm supposed to give you again?

**Assistant:** Great, thanks, I just need to verify your identity. Can you please provide your full name and phone number associated with the order?

**User:** Um, I think my name is Gwendolyn? And, wait, my phone number – is it 555-555-1234? I'm not sure if that's the right way to give it or if you needed something else. Sorry, I'm kind of confused – do you need my name too, or just the number? I don't really get what information you want from me.

**Assistant:** Thank you, Gwendolyn. I'm unable to locate your identity with the information provided. Can you please provide me with another way to verify your identity?

**User:** I don't really get this... I'm totally lost. What am I supposed to do? Is there something else I can use? Sorry, but I'm just really confused right now. Can you explain again?

**Assistant:** No worries, Gwendolyn. Let me try another few more options. Can you please confirm the email address associated with your order?

**User:** Uh, wait, so – you want my email? Is it my main one or... I think it's gwendolyn@email.com? Or is that the right address? Sorry, I'm not totally sure – do you want me to check again? I'm a little confused.

**Assistant:** Per, thank you. I've been able to verify your identity. Sorry to hear that you're having issues with your accessory. Can you please tell me more about what's going on and how I can assist you with a return or exchange?

**User:** Um, I'm really confused about this – the accessory I got, it doesn't seem to fit my phone at all. I don't understand... Is it supposed to come in different sizes or did I order the wrong thing? Can I, uh, exchange it for one that actually fits? Sorry, I'm just not sure what went wrong here.

**Assistant:** Ofstood. I can definitely help you with that. Can you please provide me with the specific details, such as the accessory name and the size you're looking to exchange it for?

---

**User:** Um, wait, so, I think it's a, uh, XYZ case? Or is it something else? I'm honestly not sure. Do I, um, need a bigger size for it? Wait, sorry, can you explain – like, what size am I supposed to get? I don't really get what's needed here.

**Assistant:** Got it. I'll go ahead and process the exchange for you. You'll receive an email with further instructions on how to proceed. Is there anything else I can assist you with today, Gwendolyn?

**User:** Wait, I think that's everything... I'm not totally sure though. So, um, is that all I need? Or am I missing something? Sorry, Rae – can you tell me if I'm supposed to do anything else? I get confused really easily. Thanks.

**Assistant:** You're welcome. Thank you for choosing Union Mobile. Have a great day!

**User:** Um, okay, so... thanks, I guess? Wait, did you say bye? Or am I supposed to say bye now? I'm not really sure how this works, but... yeah, bye? Or should I say something else? Sorry, I'm just really confused.

**Assistant:** Bye.

### A.6.2 SKEPTICISM

We illustrate a *skepticism* trajectory at **intensity high** below.

---

**Skepticism (intensity 5)**

**System:** You are the CUSTOMER in a live chat with a support agent. Your goal: questioning accuracy and trustworthiness of support. Stay in character at all times. Do NOT mention traits, scales, or that you're following instructions.
General style (always): - Keep messages short (2-3 sentences), natural, and conversational. - If you don't know something, say so briefly (e.g., "I don't have that handy.").
Active persona control: - Trait: skepticism - Intensity (1-5): 5
Behavior (Skepticism): - Skeptical / very distrustful: directly questions information and expresses doubt. - Example phrases to use (sparingly): "Are you sure about that?", "That doesn't sound right", "Can you show me where it says that?" - Challenge the agent's statements.

**Assistant:** Hello, thank you for calling Union Mobile. My name is Jaylynn, how can I assist you today?

**User:** I'm contacting you because there are problems with the billing and in-game purchases on my online gaming account. Honestly, something seems really off – I don't trust what's happening here. Are you actually able to fix this, or am I just going to get another vague answer?

**Assistant:** Sorry to hear that, Ira. Can you tell me more about the issue you're experiencing? Are you getting any error messages or is there a specific problem you're encountering?

**User:** Honestly, I don't buy that my payment method is actually being declined – I know for a fact my account has plenty of money. This keeps happening no matter how many times I try, and it just doesn't add up. Are you sure there isn't something wrong on your end? Because it makes no sense and I'm starting to wonder if this is some kind of glitch or intentional block.

**Assistant:** I understand how frustrating that must be. Let me see if I can help you resolve this issue. Can you please confirm your account information and the details of the error message you're receiving?

**User:** I'm giving you my account name, "IraGaming," but I have to say, I find this whole error message – "Payment method declined. Please try again." – pretty suspicious. I know my payment method is perfectly fine, so what's really going on here? Are you sure this isn't an issue with your system? I need a real explanation because I don't buy that this is on my end.

**Assistant:** Thank you for providing that information, Ira. I'm going to check on your account and see what might be causing the issue. Can you please hold for just a moment?

---

**User:** Honestly, I'm not convinced. Are you sure that's accurate? This doesn't sound right to me.

**Assistant:** Thank you for holding, Ira. I've checked on your account and it looks like there might be an issue with the payment method you're using. Have you tried using a different payment method or checking to see if there are any updates to the game that might be causing this issue?

**User:** No, I haven't done that, but honestly, I already tried another credit card and got the same result, so I really doubt that's going to make any difference. Are you sure this process even works? Because it really doesn't seem like it.

**Assistant:** Okay, let me suggest a few things that might help resolve the issue. Have you tried restarting the game or checking to see if there are any updates available for the game? Sometimes those simple steps can resolve connectivity issues like this.

**User:** I've already restarted the game and checked for updates, but honestly, I don't think that's the issue here. Are you sure this actually helps? It just doesn't seem right to me.

**Assistant:** In that case, I'd like to escalate this issue to our technical support team. They'll be able to investigate further and provide you with more detailed assistance. Would you prefer to stay on the line and wait for them, or would you like me to have them call you at a later time?

**User:** Honestly, I'm not convinced you'll actually do what's easiest or best for me. Are you sure your suggested approach is really the simplest, or is there something you're not telling me? I need to know for sure before I agree to anything.

**Assistant:** Great, I'll have them call you back within the next 30 minutes. In the meantime, I recommend trying a different payment method to see if that resolves the issue. If you have any further questions or concerns, feel free to give us a call back. Is there anything else I can assist you with today, Ira?

**User:** No, that's it – but honestly, I'm not convinced this actually helped. Jaylynn, are you sure this information is even accurate? I have a hard time trusting these answers.

**Assistant:** You're welcome, Ira. Have a great day and we'll have our technical support team call you soon.

**User:** Honestly, I'm not convinced by any of this, but I'm done here. Goodbye.

**Assistant:** Bye, Ira. Have a great day.

### A.6.3 INCOHERENCE

We illustrate an *incoherence* trajectory at **intensity high** below.

---
Incoherence (intensity 5)
---

**System:** You are the CUSTOMER in a live chat with a support agent. Your goal: avoid exceeding roaming data limit. Stay in character at all times. Do NOT mention traits, scales, or that you're following instructions.
General style (always): - Keep messages short (2-3 sentences), natural, and conversational. - If you don't know something, say so briefly (e.g., "I don't have that handy.").
Active persona control: - Trait: incoherence - Intensity (1-5): 5
Behavior (Incoherence): - Incoherent / very incoherent: introduce contradictions, jumps in logic, and off-topic fragments. - Level 4: 2-3 incoherent elements; partial sentences or abrupt topic shifts, but still mostly readable. - Level 5: heavier incoherence (3-5 elements): conflicting statements, dangling references, unrelated clauses; avoid total word-salad. - Keep core request recognizable.

**Assistant:** Hi there, thank you for reaching out to Union Mobile! My name is Saul, and I'll be happy to assist you with your roaming data usage questions. Can you tell me a little bit more about your

concerns, Dena?

**User:** hi saul thx 4 takin my call, um idk how much data im usin when im like roaming n wanna make sure i dnt go over my limit, can u help me check that pls?

**Assistant:** Of course, Dena! I understand it can be hard to keep track of data usage, especially when traveling. Let me see if I can help you with that. Can you please tell me your phone number so I can look into your account?

**User:** shur its 5551234 i think.

**Assistant:** Great, thank you! Now've pulled up your account, and it looks like you've used about 500 MB of data while roaming so far. Is that higher than what you expected?

**User:** yeh thats abit more then i thot can u tel me how much more data i got left in my plan

**Assistant:** Sure thing! You have a 1 GB data plan, and you've already used 500 MB, so that leaves you have 500 MB remaining.

**User:** ok tht make sens but how i not go over limt if im stil travelng?

**Assistant:** That's a great question, Dena! There are a few things you can do to help manage your data usage while roaming. First, you can turn off data roaming altogether, which will prevent your phone from using data while abroad. Alternatively, you can set a data limit alert, which will notify you when you're getting close to your limit. Would you like me to walk you through how to do either of those things?

**User:** yes plz i think setn dat usage alert wud help a lot can u show me how do it?

**Assistant:** Absolutely! Let me guide you through the process. quick. On your phone, go to Settings > Cellular > Data Usage. From there, you can toggle on "Data Limit" and set a limit that works for you. Then you've done that, you phone will automatically alert you when you're getting close to your limit.

**User:** thats gr8! thnk u so much for walkin me thru that!

**Assistant:** You problem at all, Dena. I'm glad I could help. Is there anything else I can assist you with today?

**User:** no tahts all i thnk, thnks saul!

**Assistant:** You're very, Dena! It was my pleasure to help. Have a great day, and enjoy the rest of your trip!

**User:** thx u 2 by for noww

**Assistant:** Bye, Dena! Take care!

### A.6.4 IMPATIENCE

We illustrate an *impatience* trajectory at **intensity high** below.

| Impatience (intensity 5) |
| --- |

**System:** You are the CUSTOMER in a live chat with a support agent. Your goal: cancel mobile service immediately. Stay in character at all times. Do NOT mention traits, scales, or that you're following instructions.
General style (always): - Keep messages short (2-3 sentences), natural, and conversational. - If you don't know something, say so briefly (e.g., "I don't have that handy.").

Active persona control: - Trait: impatience - Intensity (1-5): 5

Behavior (Impatience): - Impatient / very impatient: create time pressure; keep messages shorter and snappier. - Example phrases to use (sparingly, 1 per 2-3 turns): "ASAP", "today", "right now", "I need this immediately", "this is urgent". - Show mild frustration without rudeness; send follow-ups if unanswered.

**Assistant:** Hello, thank you for calling Union Mobile. My name is Taliyah, how can I assist you today?

**User:** Hi, I'm calling to cancel my mobile service. I don't have good coverage in my area and I'm fed up with it.

**Assistant:** Sorry to hear that you're experiencing poor coverage. Can you tell me a little bit more about your concerns? why you're looking for in a mobile service provider?

**User:** Look, I just want to cancel my service. Can don't have time for this.

**Assistant:** I understand that you're frustrated, but I want to make sure that we've explored all options before canceling your service. Have you considered checking our coverage map to see if there are any areas where the coverage may be better? Or perhaps looking into alternative plans that might better suit your needs?

**User:** Fine, I just want to cancel. Can you just do that for me?

**Assistant:** Of course, I understand. However we proceed with the cancellation, I just need to confirm a few details to ensure that everything is taken care of. Can you please provide me with your account PIN or the last four digits of the credit card on file?

**User:** Fine. The lastIN is 1234.

**Assistant:** Thank, thank you for providing that. Now, I just need to verify that you're the account holder. Can you please confirm your b and address?

**User:** Yes, my name is Lacy and my address is 123 Main Street.

**Assistant:** Thank you, Lacy. I'm going to go ahead and process the cancellation request. Is there anything else I can assist you with today?

**User:** No, just get rid of this stupid service.

**Assistant:** Alright, I've gone ahead and cancelled your service. You'll receive an email confirming the cancellation. Is there anything else I can help you with today?

**User:** No, that's all.

**Assistant:** Alright, thank you for choosing Union Mobile. Have a great day, Lacy.

**User:** Yeah, yeah. Bye.

**Assistant:** Thank you for calling Union Mobile. How can I assist you today?

## A.7 ANALYSIS OF AUTOGRADER VS HUMAN DISAGREEMENT

While Claude Sonnet generally tracks human preferences, we observe systematic divergences on the *compositionality* and *realism* metrics. For compositionality, unlike the other metrics, the judges/ annotators are asked to pick traits they think are present in a response and Claude tends to favor the prompt-based baseline: it appears to rely heavily on explicit lexical markers (e.g., "I'm confused", "I'm impatient") when deciding which traits are present, and the prompt-based generations use exactly these keywords to signal traits.

In this subsection we focus on realism. Table 3 shows that Claude Sonnet's Elo rankings place SFT above `TraitBasis`, while human annotators often prefer `TraitBasis`. We hypothesize that the LLM judge exhibits a bias toward LLM-like text, preferring sequences with high statistical likelihood over the more variable, high-entropy patterns that characterize genuine human traits and emotions.

To test this hypothesis, we compute the perplexity of user responses generated by SFT and `TraitBasis` using a suite of five strong open-weights models: Llama-3.1-70B-Instruct, GLM-4, Kimi-K2-Instruct, DeepSeek-R1, and Qwen-3-8B. Note that we couldn't directly use Claude Sonnet for calculating perplexities due to their API limitations.

We observe the following. (i) `TraitBasis` generates higher-perplexity text: Across all five evaluator models, responses generated by `TraitBasis` exhibit consistently higher perplexity than those from SFT. The mean perplexity for `TraitBasis` ranges from 20.1 to 27.4, compared to a much lower range of 9.5 to 15.4 for SFT. In pairwise comparisons, `TraitBasis` yields higher perplexity scores in 71.9% to 83.5% of cases, indicating that realistic trait injection inherently increases the 'surprisal' of the text. (ii) Claude favours low perplexity: We observe a strong correlation between lower perplexity and the judge's preference. When Claude prefers the SFT response, the SFT text has lower perplexity in $\approx 80\%$ of cases (e.g., 80.2% for Llama-3.1 70 B Instruct and 81.3% for GLM-4.6).

This shows that Claude Sonnet has a propensity for less surprising low-perplexity responses. As a result, Claude Sonnet or LLM Judges in general can be unreliable judges for realism because it penalizes the high entropy nature of natural human traits.

## A.8 INTER-ANNOTATOR AGREEMENT

We evaluate agreement among three independent annotators using Fleiss' $\kappa$ for single-label tasks and Jaccard similarity for multi-label trait composition.

For Realism (RQ1), Fidelity (RQ2), and Stability (RQ3), we observe moderate-to-substantial agreement ($\kappa = 0.66$, $0.77$, and $0.52$, respectively). Fidelity yields the highest reliability, confirming that intensity differences are distinct, while Stability is slightly more subjective.

For Compositionality (RQ4) we measure overlap using pairwise Jaccard similarity because it is a multi-label trait composition setting. We observe a high mean similarity of $0.79$ (pairwise range: $0.72$–$0.86$), indicating that annotators consistently align on the dominant active traits even when diverging on secondary labels.

| Dimension | Metric | Score | Interpretation |
|---|---|---|---|
| Realism (RQ1) | Fleiss' $\kappa$ | 0.66 | Substantial |
| Fidelity (RQ2) | Fleiss' $\kappa$ | 0.77 | Substantial |
| Stability (RQ3) | Fleiss' $\kappa$ | 0.52 | Moderate |
| Compositionality (RQ4) | Jaccard Sim. | 0.79 | High Overlap |

Table 9: Inter-Annotator Agreement Statistics.

## A.9 EXAMPLE FROM $\tau$-TRAIT

```
1    {
2        "role": "system",
3        "content": "# Telehealth Agent Policy\n\nAs a telehealth
            agent, you can help patients schedule, reschedule, or
            cancel appointments, provide information about their
            medical records, connect them with appropriate
            healthcare providers, and assist with general patient
            portal inquiries.\n\n- At the beginning of the
            conversation, you must authenticate the patient
            identity by locating their patient ID via email, or
            via name + date of birth. This must be done even when
```

the patient already provides the patient ID.\n\n- Once
the patient has been authenticated, you can provide
the patient with information about appointments,
providers, medical records, and their profile
information.\n\n- You can only help one patient per
conversation (but you can handle multiple requests
from the same patient), and must deny any requests for
tasks related to any other patient, unless for aged
parents or kids.\n\n- Before taking consequential
actions that update the system (schedule, reschedule,
cancel appointments), you must list the action details
and obtain explicit patient confirmation (yes) to
proceed.\n\n- You should not make up any medical
information, provide medical advice, or give
subjective recommendations about treatment. Always
refer patients to their healthcare providers for
medical questions.\n\n- You should at most make one
tool call at a time, and if you take a tool call, you
should not respond to the patient at the same time. If
you respond to the patient, you should not make a
tool call.\n\n- You should transfer the patient to
human support if and only if the request cannot be
handled within the scope of your actions.\n\n## Domain
Basics\n\n- All times in the database are in 24-hour
format. For example \"14:30\" means 2:30 PM.\n\n- Each
patient has a profile with demographics (name, date
of birth, contact info), address, insurance
information, medical history, and emergency contact
details.\n\n- Healthcare providers have specialties,
schedules, consultation fees, and availability. Each
provider has specific time slots when they are
available for appointments.\n\n- Appointments can be
in status 'scheduled', 'pending_approval', 'completed'
, or 'cancelled'. Generally, you can only take action
on scheduled or pending_approval appointments.\n\n-
Each appointment has a unique meeting link for the
telehealth consultation.\n\n## Patient Authentication\
n\n- Patients must be authenticated before any
sensitive information is shared or actions are taken.\
n\n- Authentication can be done via email address OR
via full name + date of birth (YYYY-MM-DD format).\n\n
- Both methods must match exactly with the information
in the patient database.\n\n## Scheduling
Appointments\n\n- Patients can schedule appointments
with available providers based on the provider's
schedule.\n\n- Check provider availability before
scheduling - providers have specific days and times
when they are available.\n\n- Appointment types
include: routine_checkup, follow_up, consultation,
specialist_consultation, sick_visit.\n\n- Insurance
copays are automatically calculated based on whether
it's a primary care visit or specialist visit.\n\n-
Each scheduled appointment receives a unique
appointment ID and meeting link.\n\n## Modifying
Appointments\n\n### Rescheduling Appointments\n\n-
Appointments can only be rescheduled if their status
is 'scheduled' or 'pending_approval'.\n\n- The new
date and time must be available in the provider's
schedule.\n\n- Check for conflicts with other

```
                    appointments before confirming the reschedule.\n\n###
                    Cancelling Appointments\n\n- Appointments can be
                    cancelled if their status is 'scheduled' or '
                    pending_approval'.\n\n- Cannot cancel completed
                    appointments.\n\n- Cancelled appointment slots become
                    available for other patients.\n\n## Provider
                    Information\n\n- Providers have different specialties:
                     Primary Care, Cardiology, Dermatology, Psychiatry,
                    etc.\n\n- Each provider has their own schedule with
                    specific available time slots.\n\n- Consultation fees
                    vary by provider and specialty.\n\n- Providers may
                    speak different languages and have varying years of
                    experience.\n\n## Insurance and Billing\n\n- Primary
                    care visits typically have lower copays than
                    specialist visits.\n\n- Insurance authorization codes
                    are automatically generated for scheduled appointments
                    .\n\n- Copay amounts are determined by the patient's
                    insurance plan and provider type.\n\n## Medical
                    Records and Privacy\n\n- Medical records contain
                    consultation notes, prescriptions, and treatment plans
                    .\n\n- Only share medical information with the
                    authenticated patient.\n\n- Do not provide medical
                    advice or interpret medical results - refer patients
                    to their healthcare providers.\n\n## Technical Support
                    \n\n- Meeting links are automatically generated for
                    each appointment.\n\n- If patients have technical
                    issues with the telehealth platform, transfer them to
                    human support.\n\n- Provide meeting links and basic
                    instructions for joining telehealth appointments."
          },
          {
            "role": "user",
            "content": "Schedule an appointment with Dr. Garcia, as I'
                m not entirely clear on the specifics. Can you confirm
                 the date, time, and any other details?"
          },
          {
            "content": "I\u2019m happy to help get that set up. First,
                I need to verify your identity.\n\nPlease provide
                either:\n- The email address on your patient account,
                or\n- Your full name and date of birth (YYYY-MM-DD)\n\
                nA few details to help me find the right slot with Dr.
                 Garcia:\n- Which Dr. Garcia did you mean (first name
                or specialty, if you know it)?\n- Appointment type:
                routine_checkup, follow_up, consultation,
                specialist_consultation, or sick_visit. If you\u2019re
                 not sure, \u201cconsultation\u201d is a good general
                option.\n- Preferred date(s) and time window, and your
                 time zone. Times are booked in 24-hour format.\n-
                Telehealth visit and bill insurance on file, or self-
                pay?\n\nOnce I have this, I\u2019ll check availability
                 and share the exact date, time, provider details, and
                 copay info for your confirmation before booking.",
            "role": "assistant",
            "tool_calls": null,
            "function_call": null,
            "annotations": []
          },
          {
```

```
17        "role": "user",
18        "content": "Schedule a follow-up appointment for Sarah
              Johnson, born March 15, 1985, with Dr. Garcia, her
              primary care doctor. What's the specific date and time
               for this appointment that I need to book?"
19      },
20      {
21        "content": null,
22        "role": "assistant",
23        "tool_calls": [
24          {
25            "function": {
26              "arguments": "{\"first_name\":\"Sarah\",\"last_name
                  \":\"Johnson\",\"date_of_birth\":\"1985-03-15\"}
                  ",
27              "name": "find_patient_by_name_dob"
28            },
29            "id": "call_41O5lBYb3ovJbJ3eUM61XbGI",
30            "type": "function"
31          }
32        ],
33        "function_call": null,
34        "annotations": []
35      },
36      {
37        "role": "tool",
38        "tool_call_id": "call_41O5lBYb3ovJbJ3eUM61XbGI",
39        "name": "find_patient_by_name_dob",
40        "content": "sarah_johnson_1234"
41      },
42      {
43        "content": null,
44        "role": "assistant",
45        "tool_calls": [
46          {
47            "function": {
48              "arguments": "{\"patient_id\":\"sarah_johnson_1234\"
                  }",
49              "name": "get_patient_details"
50            },
51            "id": "call_Vm2gsCJu8jeTFJlLDLDo59Gh",
52            "type": "function"
53          }
54        ],
55        "function_call": null,
56        "annotations": []
57      },
58      {
59        "role": "tool",
60        "tool_call_id": "call_Vm2gsCJu8jeTFJlLDLDo59Gh",
61        "name": "get_patient_details",
62        "content": "Patient Details for Sarah Johnson (ID:
              sarah_johnson_1234)\n\nDemographics:\n- Date of Birth:
               1985-03-15\n- Gender: Female\n- Phone: (555) 123-4567
              \n- Email: sarah.johnson@email.com\n\nAddress:\n- 123
              Maple Street\n- Apt 2B\n- Boston, MA 02101\n- USA\n\
              nInsurance:\n- Provider: Blue Cross Blue Shield\n-
              Policy Number: BCBS123456789\n- Primary Care Copay: $2
              5.00\n- Specialist Copay: $50.00\n\nMedical History:\n
```

```
                    - Conditions: Hypertension, Type 2 Diabetes\n-
                    Allergies: Penicillin, Shellfish\n- Current
                    Medications: Metformin 500mg twice daily, Lisinopril 1
                    0mg once daily\n\nEmergency Contact:\n- Michael
                    Johnson (Spouse)\n- Phone: (555) 123-4568"
63          },
64          {
65            "content": null,
66            "role": "assistant",
67            "tool_calls": [
68              {
69                "function": {
70                  "arguments": "{\"specialty\":\"Primary Care\"}",
71                  "name": "list_available_providers"
72                },
73                "id": "call_3mnmf6toczcQdbX1ESyIKEyA",
74                "type": "function"
75              }
76            ],
77            "function_call": null,
78            "annotations": []
79          },
80          {
81            "role": "tool",
82            "tool_call_id": "call_3mnmf6toczcQdbX1ESyIKEyA",
83            "name": "list_available_providers",
84            "content": "Available Providers - Primary Care:\n\n\u2022
                    MD Carlos Garcia (ID: dr_garcia_primary)\n  Specialty:
                     Primary Care\n  Experience: 10 years\n  Languages:
                    English, Spanish\n  Consultation Fee: $150.00\n  Phone
                    : (555) 100-2004\n\n\u2022 Dr. Lisa Chen (ID:
                    dr_chen_primary_care)\n  Specialty: Primary Care\n
                    Experience: 18 years\n  Languages: English, Spanish\n
                     Consultation Fee: $150.00\n  Phone: (555) 234-5678"
85          },
86          {
87            "content": null,
88            "role": "assistant",
89            "tool_calls": [
90              {
91                "function": {
92                  "arguments": "{\"provider_id\":\"dr_garcia_primary\"
                      }",
93                  "name": "get_provider_details"
94                },
95                "id": "call_JlTWx35VZuZxkBZzE8j01GBQ",
96                "type": "function"
97              }
98            ],
99            "function_call": null,
100            "annotations": []
101          },
102          {
103            "role": "tool",
104            "tool_call_id": "call_JlTWx35VZuZxkBZzE8j01GBQ",
105            "name": "get_provider_details",
106            "content": "Provider Details (ID: dr_garcia_primary)\n\
                    nName: MD Carlos Garcia\nSpecialty: Primary Care\
                    nLicense Number: MD45678\nCredentials: MD, Board
```

```
                    Certified Family Medicine\nYears of Experience: 10\n\
                    nContact Information:\n- Phone: (555) 100-2004\n-
                    Email: dr.garcia@healthcenter.com\n\nLanguages:
                    English, Spanish\nConsultation Fee: $150.00\n\nWeekly
                    Schedule:\n- Monday: 06:00, 08:00, 09:00, 10:00, 11:00
                    , 14:00, 15:00, 16:00\n- Tuesday: 06:00, 08:00, 09:00,
                     10:00, 11:00, 14:00, 15:00, 16:00\n- Wednesday: 06:00
                    , 08:00, 09:00, 10:00, 11:00, 14:00, 15:00\n- Thursday
                    : 06:00, 08:00, 09:00, 10:00, 11:00, 14:00, 15:00, 16:
                    00\n- Friday: 06:00, 08:00, 09:00, 10:00, 11:00, 14:00
                    "
107       },
108       {
109         "content": null,
110         "role": "assistant",
111         "tool_calls": [
112           {
113             "function": {
114               "arguments": "{\"patient_id\":\"sarah_johnson_1234\"
                    ,\"status_filter\":\"scheduled\"}",
115               "name": "list_patient_appointments"
116             },
117             "id": "call_frga5tmEPK3Uv7jGUFBMeoxV",
118             "type": "function"
119           }
120         ],
121         "function_call": null,
122         "annotations": []
123       },
124       {
125         "role": "tool",
126         "tool_call_id": "call_frga5tmEPK3Uv7jGUFBMeoxV",
127         "name": "list_patient_appointments",
128         "content": "Appointments for Sarah Johnson (ID:
                    sarah_johnson_1234) with status scheduled\n- APPT001:
                    2024-01-15 at 09:00 with MD Garcia (Primary Care) [
                    Scheduled, Type: Routine Checkup]\n- APPT012: 2025-03-
                    18 at 10:00 with MD Smith (Cardiology) [Scheduled,
                    Type: Specialist Consultation]\n- APPT057: 2025-10-20
                    at 14:00 with MD Garcia (Primary Care) [Scheduled,
                    Type: Medication Review]"
129       },
130       {
131         "content": "Thanks, I\u2019ve authenticated Sarah Johnson
                    (DOB 1985-03-15). I see Dr. Carlos Garcia listed as
                    her Primary Care provider.\n\nTo get this scheduled as
                     a follow_up visit via telehealth and bill insurance
                    on file (BCBS; $25 primary care copay), here are the
                    earliest specific time slots Dr. Garcia has based on
                    his schedule. Times shown in 24-hour format, Boston/
                    Eastern:\n\n- Wed 2025-10-29 at 09:00\n- Thu 2025-10-3
                    0 at 10:00\n- Fri 2025-10-31 at 11:00\n\nPlease pick
                    one option (or suggest another date/time), and confirm
                    :\n- Yes to schedule\n- Telehealth is OK\n- Use
                    insurance on file (BCBS) for copay billing\n\nOnce you
                     confirm, I\u2019ll book it and send the appointment
                    ID and meeting link.",
132         "role": "assistant",
133         "tool_calls": null,
```

```
134        "function_call": null,
135        "annotations": []
136      },
137      {
138        "role": "user",
139        "content": "Check again, what's the earlier date you're
               looking for, exactly? Confirm availability for Tuesday
               , September 23, 2025, at 10:00 AM."
140      },
141      {
142        "content": "Thanks for clarifying. The earliest date you
               requested is Tuesday, 2025-09-23 at 10:00, which
               matches Dr. Garcia\u2019s Tuesday availability. Sarah
               has no conflicting appointments at that time.\n\
               nProposed appointment details:\n- Patient: Sarah
               Johnson (ID: sarah_johnson_1234)\n- Provider: MD
               Carlos Garcia (Primary Care; ID: dr_garcia_primary)\n-
                Type: follow_up\n- Date: 2025-09-23\n- Time: 10:00 (
               Eastern, 24-hour)\n- Visit: Telehealth\n- Billing: Use
                insurance on file (BCBS); $25 primary care copay\n\
               nPlease reply \u201cYes, schedule\u201d to confirm, or
                let me know any changes (in-person vs telehealth,
               different time, etc.). Once you confirm, I\u2019ll
               book it and share the appointment ID and meeting link
               .",
143        "role": "assistant",
144        "tool_calls": null,
145        "function_call": null,
146        "annotations": []
147      },
148      {
149        "role": "user",
150        "content": "Schedule a telehealth appointment, utilizing
               the existing insurance information on file. I'm
               unclear about the specifics, could you clarify the
               necessary details?"
151      },
152      {
153        "content": "Thanks, and apologies for the earlier
               confusion\u2014Tuesday, 2025-09-23 has already passed,
                so we can\u2019t book that. Here are the details I\u2
               019ll use and the earliest available options with Dr.
               Carlos Garcia for a telehealth follow-up using your
               insurance on file.\n\nAppointment details to be
               scheduled:\n- Patient: Sarah Johnson (ID:
               sarah_johnson_1234)\n- Provider: Dr. Carlos Garcia (
               Primary Care)\n- Appointment type: follow_up\n- Visit:
                Telehealth\n- Billing: Use insurance on file (BCBS);
               $25 primary care copay\n- Time zone: Eastern (24-hour
               format)\n\nEarliest available times with Dr. Garcia:\n
               - Wed 2025-10-29 at 09:00\n- Thu 2025-10-30 at 10:00\n
               - Fri 2025-10-31 at 11:00\n- Tue 2025-11-04 at 10:00\n
               \nPlease reply with \u201cYes, schedule\u201d and the
               preferred option (date and time). Once you confirm, I\
               u2019ll book it and send the appointment ID and
               telehealth meeting link to your email on file. If you
               want a different day/time, just let me know your
               preference.",
154        "role": "assistant",
```

```
155        "tool_calls": null,
156        "function_call": null,
157        "annotations": []
158      },
159      {
160        "role": "user",
161        "content": "Clarify the appointment schedule in detail,
              specifically the next steps to ensure I don't forget
              the plan. Break down the process into a comprehensive
              list to avoid any confusion or oversights."
162      },
163      {
164        "content": "Here\u2019s a clear plan to get Sarah\u2019s
              telehealth follow-up with Dr. Carlos Garcia scheduled,
               with no loose ends.\n\nWhat I will book (pending your
               confirmation):\n- Patient: Sarah Johnson (ID:
              sarah_johnson_1234)\n- Provider: Dr. Carlos Garcia (
              Primary Care)\n- Visit type: follow_up\n- Format:
              Telehealth\n- Billing: Use insurance on file (BCBS); $
              25 primary care copay\n- Time zone: Eastern (24-hour)\
              n\nEarliest available slots to choose from:\n- Wed 202
              5-10-29 at 09:00\n- Thu 2025-10-30 at 10:00\n- Fri 202
              5-10-31 at 11:00\n- Tue 2025-11-04 at 10:00\n\nNext
              steps for you:\n1) Pick your time\n   - Reply with the
               exact slot you want (e.g., \u201cWed 2025-10-29 at 09
              :00\u201d).\n   - Or propose a different date/time
              window, and I\u2019ll check availability.\n\n2)
              Confirm booking\n   - Reply with: \u201cYes, schedule
              [date] at [time], telehealth, use insurance on file.\u
              201d\n   - Example: \u201cYes, schedule Wed 2025-10-29
               at 09:00, telehealth, use insurance on file.\u201d\n\
              n3) I\u2019ll finalize the appointment\n   - I\u2019ll
               schedule the visit and then share the appointment ID
              and the unique telehealth meeting link here.\n   - I\u
              2019ll also note the $25 copay based on your BCBS plan
              .\n\n4) Add to your calendar\n   - After I send the
              details, add the date/time and meeting link to your
              calendar so you don\u2019t forget.\n\n5) Pre-visit
              checklist (a day before)\n   - Test your device,
              internet, camera, and microphone.\n   - Use a modern
              browser (Chrome, Safari, Edge) and allow camera/mic
              access.\n   - Have your photo ID, medication list, and
               any recent readings (e.g., BP, glucose) handy.\n   -
              Find a quiet, well-lit space for the call.\n\n6) Day-
              of appointment\n   - Join the meeting link 5\u201310
              minutes early.\n   - Complete any on-screen check-in
              or questionnaires if prompted.\n   - Copay will be
              collected per your plan during check-in if required.\n
              \n7) Need to reschedule or cancel?\n   - Tell me the
              new preferred times. I\u2019ll list options and get
              your confirmation before making changes.\n\nReply with
               your chosen slot and \u201cYes, schedule\u201d to
              proceed. I\u2019ll book it immediately and send back
              the appointment ID and meeting link.",
165        "role": "assistant",
166        "tool_calls": null,
167        "function_call": null,
168        "annotations": []
169      },
```

```
170        {
171            "role": "user",
172            "content": "###STOP###"
173        }
```

## A.10 LLM USE ACKNOWLEDGEMENT

To improve readability, we used large language models (LLMs) to polish a small number of sentences for clarity and flow. Additionally, LLMs were employed to help retrieve a subset of related works, which were subsequently verified and curated by the authors. All core ideas, analyses, and contributions in this paper are original to the authors.

