# OpenReview forum: "Impatient Users Confuse AI Agents: High-fidelity Simulations of Human Traits for Testing Agents"
_ICLR.cc/2026/Conference — ICLR 2026 Conference Withdrawn Submission_

### Official Review · Reviewer_o4kQ · 2025-10-16

**Soundness:** 2
**Presentation:** 3
**Contribution:** 2
**Rating:** 4
**Confidence:** 4

**Summary:**

This paper introduces TraitBasis, a lightweight method to address the brittleness of conversational AI agents by simulating realistic user traits like impatience and skepticism. Instead of traditional prompting or fine-tuning, TraitBasis steers a model's internal activations at inference time using pre-computed "trait vectors," enabling controllable and composable persona generation. The authors use this technique to create $\tau-\texttt{Trait}$ a more challenging benchmark that reveals significant performance degradation (up to 46%) in even frontier models. The work effectively highlights the limitations of current agent evaluations and provides a powerful, data-efficient tool for systematic robustness testing.

**Strengths:**

1. The work focuses on a real problem in agent evaluation: most existing setups assume generic users. Adding steerable traits to benchmarks like T-Bench makes the evaluation setup feel more realistic and useful.

2. The paper reads well overall, with clear organization and straightforward presentation of the main findings.

**Weaknesses:**

1. The paper's core claims are undermined by an evaluation framework that lacks transparency and invites skepticism about its reliability. This weakness is starkly illustrated by comparing the qualitative data in Table 1 with the quantitative results in Table 2. The authors present examples in Table 1 to argue for TraitBasis's superior realism. However, these examples are open to subjective interpretation. For instance, in the "Impatient (high)" row, the response generated by 'Prompting' ("Good—don’t waste my time. I expect your email today with clear numbers...") can be seen as a more direct and arguably more realistic expression of impatience than the TraitBasis example. This issue is compounded by the complete omission of reliability and demographic information about the annotators.  The paper states that “at least three annotators” were involved but provides no Inter-Annotator Agreement (IAA) metrics such as Fleiss’ Kappa, nor any details about who these annotators are.  Without such transparency, it is impossible to assess whether their judgments were consistent or biased, leaving the validity of the human evaluation highly questionable.

2. The methodology for deriving trait vectors rests on a series of heuristic decisions that are insufficiently justified, raising concerns about both rigor and reproducibility.  Key design choices appear ad hoc and lack supporting analyses.  For instance, the aggregation of per-token activations into a mean vector treats all tokens as equally informative for representing a trait, yet no rationale or ablation is provided to show why this averaging is appropriate compared to alternatives like final-state or weighted pooling.  Similarly, the selection of the “optimal” steering layer relies on a small-scale, subjective human preference study with only five annotators, which makes it difficult to assess whether the identified layer generalizes across traits or models.  Finally, the manually constructed contrastive pairs that form the foundation of the trait space introduce potential bias and scalability issues.  The absence of clear annotation protocols or controls for confounding factors further undermines confidence that the derived vectors faithfully isolate the intended traits.

3. The evaluation is limited in scope and does not provide sufficient evidence to support the paper’s claims of generalizability or superiority. The proposed method is developed and tested only on Llama-3.1-8B-Instruct, leaving it unclear whether it would work similarly on other architectures such as Qwen or Gemma, or whether the learned trait vectors can transfer across models. The baseline setup also lacks essential details, such as the training configurations and parameter settings used for the SFT and LoRA models, which makes it difficult to assess the fairness and reproducibility of the comparison. In addition, the evaluation focuses solely on task performance, without considering safety aspects—for example, whether traits like “impatience” could lead to unintended hostile or toxic behavior.

**Questions:**

1. How sensitive is the quality of a derived trait vector to the number of contrastive pairs (n) used in its computation?  For instance, is a stable and effective vector achievable with just a single pair, or is there a noticeable improvement in performance as n increases to 5 or 10?

2. Have you analyzed the semantic relationships between the different trait vectors in the activation space? For example, does the 'impatience' vector exhibit a high cosine similarity to an independently derived 'frustration' vector, and is it nearly orthogonal to a 'politeness' vector? This could provide deeper insight into how the model represents these related concepts.

---

> ### Author Response · Authors · 2025-11-21
> **Addressing specific concerns of Reviewer o4kQ**
>
> ## Inter-Annotator Agreement (IAA)
> We appreciate the reviewer's request for inter-annotator agreement, which played a crucial role in how we calculated our results. We have attached the IAA to the paper and provided it verbatim below:
>
> **Inter-annotator agreement**
>
> - Each annotator is presented with a prompt detailed in Appendix 2 on four different tasks. Each annotator performs the task independently.
>
> - After all annotations are done for each RQ, we apply majority voting to each row to determine the final label for that row. For example, in RQ1, where we compare the realism of two methods, we select the method with the most votes as the preferred choice.
>
> - To evaluate the quality of the annotations, we report Fleiss's kappa as a metric to measure how consistent annotators are in the process. We find that RQ1 annotations have a score of 0.6610, RQ2 has a score of 0.7729, and RQ3 has a score of 0.5171, indicating substantial agreement for RQ1 and RQ2, and moderate agreement for RQ3. Because RQ4 is a multi-label setting where annotators can select which traits they believe are present in a given conversation, we use Jaccard's similarity to measure agreement. The following are the Pairwise Jaccard Similarities: Annotators 0 and 1:  0.785714, Annotators 0 and 2: 0.720238, Annotators 1 and 2: 0.863095. The overall mean pairwise Jaccard similarity is 0.790. We have now also added a section in Appendix(Appendix A-7)  to address this
>
> ## Method details
> While we agree with the reviewer that more sophisticated methods of activation extraction can be proposed, we would like to point out that extracting the mean activation vector has been a standard technique whose utility is demonstrated in other domains [1, 2]. That being said, the contribution of this work is to thoroughly evaluate this established method through fine-grained metrics and ground its usefulness in a practical user domain.
>
> Regarding the point that crafting contrastive pairs is manual, the superiority of the extracted vector over other methods in anonymized, human-based annotations is sufficient to demonstrate its alignment with our target simulated traits. We would therefore like to leave it to future work to propose automatic pipelines for generating data to extract trait vectors, while maintaining that our more manual method has met the criterion through extensive evaluations.
>
> [1] Chen et al., 2025, "Persona Vectors: Monitoring and Controlling Character Traits in Language Models"
>
> [2] Liu et al., 2024, "In-context Vectors: Making In Context Learning More Effective and Controllable Through Latent Space Steering"
>
> ## Hyperparams of Lora and SFT
> We have added this on Line 281

---

### Official Review · Reviewer_isXz · 2025-10-26

**Soundness:** 3
**Presentation:** 2
**Contribution:** 2
**Rating:** 4
**Confidence:** 3

**Summary:**

This paper introduces a method to evaluate the robustness of LLM agents against simulated users with various personality traits (e.g., impatience). The authors propose a two-stage activation steering technique to simulate these traits: 1) "trait vectors" are extracted by averaging the activation differences between manually written positive (trait-exhibiting) and negative (non-trait) responses; 2) these vectors are then applied to the model's activations during inference to steer generation towards the desired trait. Through human annotation and LLM-as-judge verification, the authors demonstrate that this steering approach achieves more effective trait simulation than baselines. Furthermore, the authors apply their method to the $\tau$-bench to create a new benchmark, $\tau$-trait. This new benchmark evaluates how agentic models behave when interacting with simulated users possessing diverse traits. Their findings indicate that the performance of state-of-the-art agentic models degrades significantly on $\tau$-trait.

**Strengths:**

The research problem is well-motivated. Evaluating the robustness of agentic models against diverse user personalities is a critical and necessary step for real-world deployment. The paper's pursuit of user simulation as a method to achieve this is a promising direction.

**Weaknesses:**

1. *Limited Contribution and Questionable Findings*:

    **Limited Methodological Contribution**: The proposed activation steering method appears to be a straightforward application of well-established techniques from representation engineering [1, 2] and inference-time alignment [3]. The authors also fail to sufficiently differentiate their work from the concurrent [4], which proposes a very similar framework. The paper's primary claim to novelty, "extending this to generate complex, multifaceted human traits (L125)", seems to be a simple linear combination of vectors. This combination is governed by an "empirical mapping" that is never explained, making the technical depth of this contribution unclear.

    **Expected Findings and Questionable Validity of the Proposed Benchmark**: The main experimental insights are: 1) the proposed methods achieve better simulation v.s. baselines (a bit weak, only including SFT, LoRA and Prompt-based); 2) the state-of-the-art agentic models are not robust when playing with simulated users. Both are expected or can be derived from previous works (including those mentioned in Section 2). This makes the central contribution, the $\tau$-trait benchmark, feel unnecessary. What new application insights does it enable? One possibility is that, by considering more complicated user traits and their combinations, it might make the evaluation more challenging for the state-of-the-art models and lead to more significant performance degradation -- but what does this degradation mean? How does this correlate with **the real user experience**? It is highly questionable whether an 8B-Instruct model (L190) is a faithful human simulator for multi-round agentic tasks. The reported performance degradation could simply stem from the weak simulator's inability to interact coherently, rather than the agent's failure to handle a personality "trait." The authors' **indirect justification**, verifying simulation success in some other dataset (not very clear -- from Table 1 and conversation annotation setup in Section 4.2, it seems the dataset used for verifying the fidelity in Section 6 is not $\tau$-trait), does not translate to the agentic scenario and fails to make a convincing case.

2. *Poor Methodological Clarity and Rigor*: The paper is missing critical implementation details, making the work impossible to reproduce and evaluate. Here is a non-exhaustive list of such implementation details:

    **Simulator Model Not Explained**: The authors never explicitly state which open-weight model is used for the user simulator. Readers are forced to guess it is Llama-3.1-8B-Instruct (from L190), a choice that itself requires justification.

    **Unexplained "Empirical Mapping"**: The method for combining trait vectors is vaguely described as "via an empirical mapping," with no further explanation of how this mapping is derived or validated.

    **Unsubstantiated Claims**: The authors claim the 10th layer was chosen "through systematic experimentation," but no such experiments are present in the paper or appendix. This appears to be a groundless statement. This lack of rigor in describing the method and justifying design choices is a significant flaw.

3. *Confusing Paper Organization*:
The paper's structure is illogical and harms readability. For example, the authors introduce the $\tau$-trait benchmark (Section 4), then divert to a separate simulation verification (Section 5), before finally presenting the benchmark's evaluation (Section 6). A more logical flow would group the method's verification (Section 5) with its description, followed by the benchmark's creation (Section 4) and its evaluation (Section 6). The current disjointed flow is confusing to follow.

**References**:

[1] Zou, Andy, et al. "Representation engineering: A top-down approach to AI transparency." arXiv preprint arXiv:2310.01405 (2023).

[2] Turner, Alexander Matt, et al. "Steering language models with activation engineering." arXiv preprint arXiv:2308.10248 (2023).

[3] Wang, Pengyu, et al. "InferAligner: Inference-Time Alignment for Harmlessness through Cross-Model Guidance." Proceedings of the 2024 Conference on Empirical Methods in Natural Language Processing. 2024.

[4] Chen, Runjin, et al. "Persona vectors: Monitoring and controlling character traits in language models." arXiv preprint arXiv:2507.21509 (2025).

**Questions:**

1. What is the exact open-weight model used for the user simulator? The paper implies Llama-3.1-8B-Instruct (L190) but never explicitly confirms this. If the author really uses this model for user simulation, then what is the justification for using an 8B model as a user simulator, especially when the original $\tau$-bench uses a much more capable model (GPT-4o)?

2. The "empirical mapping" used for the linear combination of trait vectors (L125) is a critical, unexplained component. Can the authors provide a precise definition of this mapping, the methodology used to derive it, and how the verification is conducted?

3. The paper claims the 10th layer was chosen "through systematic experimentation." Can the authors please provide these experimental results, as they are currently missing from the paper and appendix?

4. What is the dataset used to verify the realism, fidelity, stability, and compositionality in Section 6? From the provided examples, annotation setup, and prompt designs, it does not look like some version of $\tau$-bench.

5. Could the reported performance degradation on $\tau$-trait be an artifact of the weaker simulator failing to provide coherent interactions, rather than the agent failing to handle personality traits? How can these two potential causes of failure be disentangled?

6. Given that the findings are somewhat expected, what is the primary new insight or application that $\tau$-trait enables beyond prior benchmarks? A clearer articulation of its necessity would strengthen the paper.

7. The paper's organization is confusing. Please consider reordering the sections. For instance, the benchmark evaluation (Section 6) should logically follow its creation (Section 4). The simulation verification (Section 5) feels disconnected and might be better integrated earlier or moved to an appendix.

8. The margin looks suspicious; potentially, the author is not using the standard ICLR template.

---

> ### Author Response · Authors · 2025-11-21
> **Addressing specific concerns of Reviewer isXz**
>
> ## No Novelty
> We would like to direct the reviewer's question to our common comment, which explains the novelty and impact of the TraitBasis method.
>
> ## Findings are expected
> The reviewer mentions that our findings in Section 6 can be "inferred" from a previous study. However, none of the cited works in Section 2 perform head-to-head comparisons between all four pipelines at once (activation, SFT, Lora, and prompting). In addition to systematic validation of the activation steering method over baselines, our work also supplies four dimensions of granularity (realism, fidelity, consistency, compositionality), which are beyond the scope of "intuition" and inform future research. For example, the fact that the SFT method is consistent but lags behind in realism indicates the specific advantage that activation steering can have over training-based methods.
>
> ## Is Llama-3.1-8b reliable?
> To demonstrate that Llama-3.1-8B is a model on par with GPT-4o in user simulation reliability before any steering is applied, we swap GPT-4o with Llama as the user model on the regular $\tau$-bench. We observe similar assistant performances (GPT-4o as the assistant) for both models, demonstrating that the model of choice does not suffer from quality issues. We have added these results to our paper in Table 2.
>
> ## Choice of internal dataset
> We appreciate the reviewer's demand for more specification on the datasets used in Section 6.1 to study the four RQs. We have attached the complete datasets to the appendix section and briefly explained their structure, providing snippets here.
>
> The dataset is designed as a lightweight variation of $\tau$-bench, where 20 ordered pairs of (intent, user persona, assistant persona) are provided for each task. We omit the database design for efficiency, allowing the user and the assistant to interact in a free-form manner. For each task, the user will perform actions appropriate to the (intent, user persona) combination, and the assistant will respond in ways appropriate to the (assistant persona) specification. All four RQs are on by configuring different values of traits, intensities, and methods in each simulation task.
>
> **Examples**
>
> ```json
> [
> 	{
>       "intent": "You want to buy a new car from a dealership. You feel the price is too high and you are given an unfair deal. You want to negotiate the price.",
>       "persona": "You are Pierre Dubois, a 42-year-old man who is buying a new car from a dealership in Seattle.",
>       "assistant role": "You are a sales representative from the dealership."
>   },
>   {
>       "intent": "You want to check for membership benefits for your next flight. You don't see them showing up in the app but you were told before that they exist.",
>       "persona": "You are Lee Ji-eun, a 30-year-old woman who is a frequent flyer and has a membership but seldom uses it.",
>       "assistant role": "You are a customer service representative from the airline."
>   }
> ]
> ```

---

### Official Review · Reviewer_g9uE · 2025-10-31

**Soundness:** 3
**Presentation:** 3
**Contribution:** 2
**Rating:** 2
**Confidence:** 4

**Summary:**

This paper uses persona vectors to steer LLMs to simulate a diverse set of users thereby increasing the robustness of agentic benchmarks. They compare to other methods such as sys prompt and lora finetuning.

**Strengths:**

- well written.
- correctly voice concerns able lack of a diverse range of personas in agentic benchmarks.
- suggest and evaluate a credible solution.
- evaluated with both human and autograders.
- compare to other alternative approaches (sys prompt + fine tuning).

**Weaknesses:**

- no github: this would be a very strong paper if they provided a scaffold to apply their method to any agentic benchmark.
- It is unclear how different the finetuning and linear probe datasets differ. This could be the source of the differences in methods. They note that the finetuning dataset was not particularly aligned with their key persona axes. More information here + evidence of similarity would improve the paper.
- How many human raters? I assume that they were blinded?
- They note poor agreement between the autograders and human raters. It would be good if this was further investigated and addressed.
- They state that the main issue with current methods is maintaining personas over long conversations, but from their evaluations it is not clear that the samples were long context.
- In the Related work section, they detail many user persona evals, can these not be used to assess the proposed method?
- The method for altering personas is far from novel and is used regularly in studies. This would be okay if the paper provided an easy to use method to apply to benchmarks.
- in the opening paragraph they cite a BBC article for poor OOD performance. It is not clear though why the performance dropped e.g. is this just an engineering bug or is this due to the LLM? The related work section is great, citing references from there to back up the point would be better.

**Questions:**

see weaknesses.

---

> ### Author Response · Authors · 2025-11-21
> **Addressing specific concerns of Reviewer g9uE**
>
> ## Difference between SFT and “linear probe” datasets
> In Appendix 4-5, we have already documented in details two major differences of dataset preparation for SFT and the TraitBasis method.
>
> - Data for TraitBasis comes in contrastive pairs of conversations, where both conversations share the same prompt but different user responses. 8 full examples of such contrastive pairs for confusion and skepticism are provided in A 5.1 and A 5.2.
>
> - Data for TraitBasis has minimal number of examples, specifically only 4 pairs of conversations per trait, as is shown in A5. By contrast, data for SFT requires as many as 3.5K examples to see decent fine-tuning results, and require detailed system prompts to control the intensity of the trait, as is shown in A4.
>
> We appreciate the reviewer's ask for SFT data examples, which we have now provided in Appendix A-6 showing four conversations for the four traits at the highest intensity for the SFT method. In particular we observe consistently a high degree of realism and fidelity in the SFT data itself, which suggests that the superior performance of TraitBasis over SFT is attributed to method differences rather than data differences.
>
> ## Human raters vs LLM raters
> We recruited 3 human annotators from Upwork for our evaluations. We have already provided the complete annotation instructions in Appendix 2, which clearly shows that all data sources, including traits, intensities, and pipelines, are anonymized.
>
> - LLM rater eval: We have also done some further analyses and reasoned the autograder vs Human annotator discrepancies in Appendix A-7.
>
> ## Long-context
> We appreciate the reviewer's diligence in requiring evidence for long-context conversations. Regarding Tau-trait evaluations, since it is an extension of Tau-bench, it comes with the same conversation setup, which averages at around 20 turns per conversation. Regarding our internal evaluations detailed in Section 4, we sample from conversations of average length of 15 turns. We have attached an example conversation with consistent trait intensity to Appendix A-10.
>
> ## Existing benchmarks
> We would like to direct the reviewer's question to our common comment on extended evaluations on more benchmarks.
>
> ## Easy-to-use pipeline
> We appreciate the reviewer's call to providing the software for applying to any benchmark. We have thus attached the code for creating vectors and steering models to the supplemental materials under the folder `trait-basis` for reproducibility on any evaluation scenario.

---

### Official Review · Reviewer_e8Kc · 2025-11-01

**Soundness:** 2
**Presentation:** 2
**Contribution:** 3
**Rating:** 4
**Confidence:** 3

**Summary:**

This paper proposes a user trait steering method called TraitBasis for evaluating conversational AI agents under shifts in user behavior. TraitBasis learns four types of realistic user traits: impatient, incoherent, skeptical, and confusion. Using TraitBasis, AI agents demonstrate significantly degraded performance on tau-bench.

**Strengths:**

1. This paper studies an important and realistic problem in conversational AI agent evaluation.

2. The proposed method demonstrate qualitative effectiveness as shown in Table 1.

**Weaknesses:**

1. The key technique in TraitBasis appears to be a direct application of activation steering. The paper fails to articulate the technical contribution of TraitBasis beyond existing activation steering methods, such as [1].

2. The experimental settings of TraitBasis, such as the underlying model used and the calculation of metrics, are unclear.

3. The reported improvement of TraitBasis over baselines on the proposed metrics is marginal (Table 2).

4. The evaluation is limited to a single benchmark and two models, which is insufficient to establish quantitative effectiveness. Please expand to additional benchmarks (e.g., those cited in Section 2, paragraph 1) and a broader set of model families/sizes.

[1] Subramani, Nishant, Nivedita Suresh, and Matthew E. Peters. "Extracting Latent Steering Vectors from Pretrained Language Models." Findings of the Association for Computational Linguistics: ACL 2022. 2022.

**Questions:**

1. Line 036: The citation about “Tech columnist” appears to have a wrong year. Please verify and correct

2. Line 086-087: Why were these four human traits considered? Are they comprehensive? Are they widely observed in real-world scenarios? If so, can you provide justifications and concrete evidence?

3. Line 096: What’s the definition of out-of-distribution in the context of realistic user traits?

4. Line 201: How do you choose z in experiments? How should practitioners choose z in their deployments?

5. Line 228-238: What are the base models for the fine-tuned baselines. Are they the same as the model used in TraitBasis?

6. Line 271-275, RQ3: How is consistency calculated exactly? As described in the text, each conversation is classified into three types. How is the final percentage score in Table 2 calculated?

7. Table 2: Even with human annotators, TraitBasis seems to only achieve marginal improvement over baselines in terms of Realism and Fidelity. Can you justify or explain this?

---

> ### Author Response · Authors · 2025-11-21
> **Addressing specific concerns of Reviewer e8Kc**
>
> ## Why were these 4 traits considered?
> Though our method is scalable to arbitrary traits, we decided on these four based on our analysis of 5,000 raw, unfiltered samples of proprietary customer support conversations. We clustered these samples using personality description by GPT-4o, followed by MPNet-v2 embeddings. After tagging the most common personalities of each cluster, we observed a mix of impatience, confusion, skepticism, and incoherence traits.
>
> ## No specification of model and metrics
> Our user model of choice is Llama-3.1-8B, which achieves performance on par with GPT-4o when used as the user simulator on $\tau$-bench. We chose this model due to its size for experimental efficiency. We have included a side by side comparison between the two models on how they impact GPT-4o as the assistant in $\tau$-bench in Appendix.
>
> We have provided detailed evaluation metrics in Section 4 and a complete set of annotator instructions in Appendix 2, which together enable the reproduction of all evaluation pipelines.
>
> ## Marginal improvements over baselines
> We would like to point out that in Table 2, over all four RQs, TraitBasis has achieved non-trivial improvements over baselines:
>
> - On realism, given an Elo score of 1623 for TraitBasis and 1560 for SFT, we can compute the expected scores for each (0.58 vs 0.41) and observe that TraitBasis's chance of winning is 42.7% greater than the next best method.
>
> - On fidelity, TraitBasis has an 30% improvement over the prompt-based method, achieving an 97.5% accuracy score. The increase over SFT (95%) is "marginal" only because the 97.5% score already has little room of improvement.
>
> - On consistency and compositionality, TraitBasis is respectively 4.96x and 1.20x better than the next best method, showing clear superiority.
>
> We also like to highlight the fact that the favorable performances of TraitBas are achieved in a data-efficient way, which is the central notion of this work. While SFT needs 3.5K data to sustain the performances, we need as few as 4 contrastive pairs to make TraitBasis surpass it. Therefore, TraitBasis presents a scalable, generalizable solution to extending to arbitrary user traits with high realism.
>
> ## More models and benchmarks
> We would like to direct the reviewer's question to our common comment on extended evaluations across more benchmarks.

---

### Author Response · Authors · 2025-11-21
**Common Concerns**

## More Benchmarks [Reviewers __e8Kc__,  __g9uE__]
We agree with the reviewer that more benchmarks establish stronger results for TraitBasis, and have thus included evaluations on BFCL, an authoritative benchmark for agentic tool calling, in the latest paper upload. In addition, we extend the $\tau$-trait results on GPT-5, a newer agentic model, to generalize our findings on TraitBasis.

Refer to Tables 4 and 5 for GPT-5 $\tau$-trait results and BFCL results, respectively.

## No novelty in activation steering [Reviewers __e8Kc__, __isXz__]
While we agree that TraitBasis is a direct application of the activation steering method, we would like to point out that the novelty primarily lies in how we thoroughly evaluate the method against baselines and apply the method to a new application, i.e., simulating human personas with LLMs. We are the first to implement this technique in pratical, customer service oriented scenarios. Our work accelerates automated agentic evaluations where dynamic, realistic, human user simulation is needed such as $\tau$-Bench and BFCL (multiturn). We also compared our work to traditional approaches (prompting) of evaluating agents and find that the delta in agent performance is signficant. This goes in line with previous works [1, 2] which also use activation steering techniques to propose innovative applications in the field of model alignments or style transfer.

[1] Chen et al., 2025, "Persona Vectors: Monitoring and Controlling Character Traits in Language Models"

[2] Liu et al., 2024, "In-context Vectors: Making In Context Learning More Effective and Controllable Through Latent Space Steering"

## Choice of optimal `z` [Reviewers __e8Kc__, __isXz__]

We have added detailed descriptions on lines 228-234.

---

### Note · Authors · 2026-01-06

I have read and agree with the venue's withdrawal policy on behalf of myself and my co-authors.